# Organic magnetic nanoparticles catalyze CO$_2$ capture in hydrogen-bonded nanocages via water-driven crystallization

Tian Wang[1,2], Aliakbar Hassanpouryouzband [3] ✉, Mengge Fan[4], Chalachew Mebrahtu [2], Lunxiang Zhang [1] ✉ & Yongchen Song [1] ✉

Limiting global warming increasingly relies on the development of environmentally friendly CO$_2$ capture strategies. Crystallization is renowned for versatile separation and purification, yet traditional compound crystallization-based CO$_2$ capture still necessitates intricate preparation processes, stringent reaction conditions, and high regenerative energy consumption. As an ambitious sustainability goal, natural water could be used as a precursor of crystallization to construct hydrogen-bonded water cages for CO$_2$ capture, but main obstacles are slow crystallization kinetics and low capture capacity. Here, a water-activation-induced crystallization strategy by organic magnetic nanoparticles (Methionine@Fe$_3$O$_4$) has been proposed for efficient CO$_2$ capture. Local water ordering strengthened by hydrophobic amino acids and abundant nucleation sites provided by nanoparticles create hotspots for hydration phase transition and crystal growth, with a CO$_2$ capture capacity of 118.7 v/v (22.7 wt%). Favorable biocompatibility and stable performance are conducive to the industrial application of this nanomaterial, and the excellent magnetic recyclable property enables simple separation from clean water. This strategy demonstrates an extraordinary CO$_2$ capture potential compared to state-of-the-art systems, thus providing an inspiration for sustainable CO$_2$ capture and storage with zero resource depletion (ZRD).

CO$_2$ capture technology has received widespread attention and development in response to climate change and environmental issues attributed by overuse of fossil energy[1–3]. Currently, liquid amine absorption is the most mature method among numerous CO$_2$ capture strategies, but many problems such as high capital, high energy intensity, toxicity and difficulty of storage still restrict its large-scale deployment[4]. Amine-containing solids with high adsorption capacity and tunable structures are effective in mitigating volatile amine losses

and equipment corrosion, but oxidative degradation of amines makes them unresponsive to sustained CO$_2$ capture cycles[5]. For the consideration of energy conservation and environmental protection, researchers have turned to the development of porous materials with high-surface area such as zeolites[6,7], porous carbons[8–10], metal-organic frameworks (MOFs)[11–13], covalent triazine frameworks (CTFs)[14], and covalent-organic frameworks (COFs)[15] for CO$_2$ capture through physical adsorption. The development of porous materials with high

[1]Key Laboratory of Ocean Energy Utilization and Energy Conservation of the Ministry of Education, School of Energy and Power Engineering, Dalian University of Technology, Dalian 116024, China. [2]Chair of Heterogeneous Catalysis and Technical Chemistry, Institute for Technical and Macromolecular Chemistry (ITMC), RWTH Aachen University, Worringerweg 2, 52074 Aachen, Germany. [3]School of Geosciences, University of Edinburgh, Grant Institute, West Main Road, Edinburgh EH9 3FE, UK. [4]School of Environmental Science and Engineering, Guangdong Provincial Key Laboratory of Environmental Pollution Control and Remediation Technology, Sun Yat-sen University, Guangzhou 510275, China. ✉e-mail: Hssnpr@ed.ac.uk; lunxiangzhang@dlut.edu.cn; songyc@dlut.edu.cn

adsorption capacity and selectivity, low heat of adsorption, good chemical and thermal stability is the key to achieve sustainable $CO_2$ capture[16]. $CO_2$ capture capacity of the composites of zeolite 5A and MOF-74 reached 7.1 mmol/g (23.8 wt%) at 1.1 bar and 288 K[17]. However, fine material design, multiple processes and stringent reaction conditions are necessary for the preparation of high-performance porous materials, such as template guidance, charring and template removal usually required for porous carbon with large pore volumes, and days-long crystal growth required for some MOFs preparation[18,19]. In addition, the competition between parasitic molecules (such as water) and $CO_2$ will ultimately deteriorate the recyclability and capacity of $CO_2$ capture in practical scenarios[20,21].

Compared to gas adsorption, crystallization-based $CO_2$ capture strategies require simple separation procedures and recycled materials, helping to reduce costs, improve efficiency and make carbon capture technology practical. $CO_2$ spontaneously nucleates and precipitates in the form of carbonate crystals in alkaline solutions, one of the most classic methods of capturing $CO_2$ via crystallization, which is known for its simple operation and controllable product[22]. Unfortunately, the economic costs increased by high energy consumption of $CO_2$ regeneration, strong corrosiveness of alkaline solutions, and non-recyclability of absorbers are prohibitive[23,24]. As shown in Supplementary Table 1, reported crystalline precursors such as hydroquinone, nitramine, ionic liquids faced severe challenges such as weak $CO_2$ capture performance, high energy input, and harsh regeneration conditions (353–433 K). Based on the moderate strength of hydrogen bond, hydrogen-bond assisted super-molecular assembly showed excellent structural flexibility and could be ideal candidate for reversible capture of guest molecules under mild condition[25-27]. Benefiting from strong hydrogen-bond donor ability and selectivity for oxyanions, guanidine compounds capture carbonate ions through electrostatic and hydrogen bonding to crystallize into hydrated carbonate salts has been reported as an alternative carbon capture method[28-31]. Although the energy required for glyoxal-bis-iminoguanidine (GBIG) sorbent regeneration was 24% lower than that of benchmark industrial $CO_2$ sorbent (monoethanolamine), its regeneration temperature was still as high as 120 °C, which could not completely avoid energy-intensive processes[28]. Inspired by the crystalline three-dimensional hydrogen-bonded framework, guanidinium sulfate (Gua$_2$SO$_4$) was used to co-crystallize with $CO_2$ to form a stable $CO_2$@Gua$_2$SO$_4$ clathrate powder, and $CO_2$ was trapped in a framework constructed by hydrogen bonds and electrostatic interactions between guanidine cations and sulfate[32,33]. The most attractive feature of the dynamic hydrogen-bond framework for capturing $CO_2$ was the low energy input

for absorbent regeneration and $CO_2$ storage capability with a gas compression ratio of ~60[33].

As an ambitious sustainability goal, it is attractive that directly constructing hydrogen-bonded cages to capture $CO_2$ via liquid water avoids of environmentally unfriendly complex compounds (Fig. 1)[34,35]. Crystalline hydrogen-bonded water cages store 160-180 m³ $CO_2$ in 1 m³ clathrate make this green $CO_2$ capture strategy more promising[36]. Based on a comprehensive comparison of several representative $CO_2$ capture methods (Supplementary Table. 2), the $CO_2$ capture strategy based on a hydrogen-bonded water cage has many advantages in addition to high storage density[35,37]: (a) insensitivity to impurities in industrial gases; (b) easy solution regeneration achieved by simple heat exchange with the environment; (c) low energy consumption for sustainable capture[35]. However, the long induction period of clathrate nucleation and slow kinetics of $CO_2$@Water crystal growth provide primary impediments to generalizing the technique on the carbon capture side[38,39]. Surfactants and nanomaterials have been proven to overcome barriers of gas-liquid diffusion and crystal nucleation by improving heat and mass transfer conditions[40]. Sulfonate-based surfactants, including sodium dodecyl sulfate (SDS), alkyl diphenyl ether bisulfonate, and sodium polystyrenesulfonate, can promote $CO_2$@Water crystallization by reducing the interfacial tension of the liquid, but the loss of surfactant due to excessive foaming of solution during crystal decomposition is regrettable[41]. Porous MOFs represented by HKUST-1 have shown excellent promotion performance on clathrate crystallization by increasing the surface area of gas-to-water contact[42]. Yet the potential cost and hazards to human and the environment are serious obstacles to the commercialization of these chemical additives. Proposing an environmentally friendly, durable, robust, and resource-saving scheme is the key to generalizing the green crystallization reaction process of $CO_2$ capture based on hydrogen-bonded water cages.

Herein, a crystallization-based ZRD scheme combined high storage density, environmental friendliness and energy efficiency is formulated for sustainable $CO_2$ capture via reversible phase transition. Organic magnetic nanoparticles (Methionine@Fe$_3$O$_4$ nanoparticles-MFNs) are designed to induce water molecules to construct hydrogen-bonded nanocages for $CO_2$ fixation via strengthened liquid activation. The local water ordering induced by hydrophobic methionine loaded on the surface of Fe$_3$O$_4$ and the abundant nucleation sites provided by nanoparticles create hotspots for hydration phase transition and crystal growth. $CO_2$ molecules are fixed in hydrogen-bonded water cages, $CO_2$ capture capacity in $CO_2$@Water crystal is determined to be 118.7 v/v (22.7 wt%), revealing its great application potential for $CO_2$

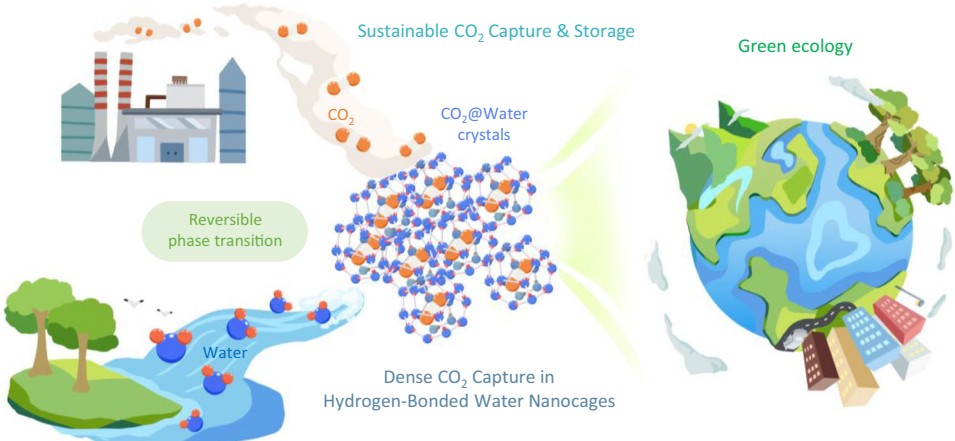

**Fig. 1 | Concept and scenario for $CO_2$ capture via hydrogen-bonded water nanocages.** Reversible conversion of $CO_2$ and water into $CO_2$@Water crystals enables dense $CO_2$ capture in hydrogen-bonded water nanocages, contributing to the development of green ecology with a sustainable $CO_2$ capture and storage scheme.

capture under practical conditions. Organic magnetic nanoparticles do not participate in co-crystallization of $CO_2$ and water, and $CO_2$ release and nanofluid regeneration can be achieved at room temperature as clathrate crystals decomposition. MFNs exhibit excellent reusability and stability in cycling experiments and negligible cytotoxicity in cell culture media, making them suitable for large-scale application. Moreover, good magnetic properties of MFNs allow for easy solid-liquid separation, the complete recovery of iron and water supports $CO_2$ capture with ZRD. The green catalysis of water crystallization by this high-performance, stable, recyclable and non-toxic material makes sustainable $CO_2$ capture feasible.

## Results

### Fabrication and characterization of MFNs

The fabrication process of MFNs by single pot co-precipitation method is schematically shown in Fig. 2a. Scanning electron microscopy (SEM) characterization on MFNs in Fig. 2b shows uniform nanosphere cluster-like shapes with dispersed particle distribution. The pitted surfaces of MFNs in the shape of spherical clusters could provide suitable nucleation sites for $CO_2$@Water clathrates crystallization[43]. Fig. 2c shows that MFNs were predominantly distributed in the range of 40–70 nm with a median size of ~54.9 nm, $Fe_3O_4$ cores distribute randomly within methionine substrate. As depicted in Fig. 2d, a thin

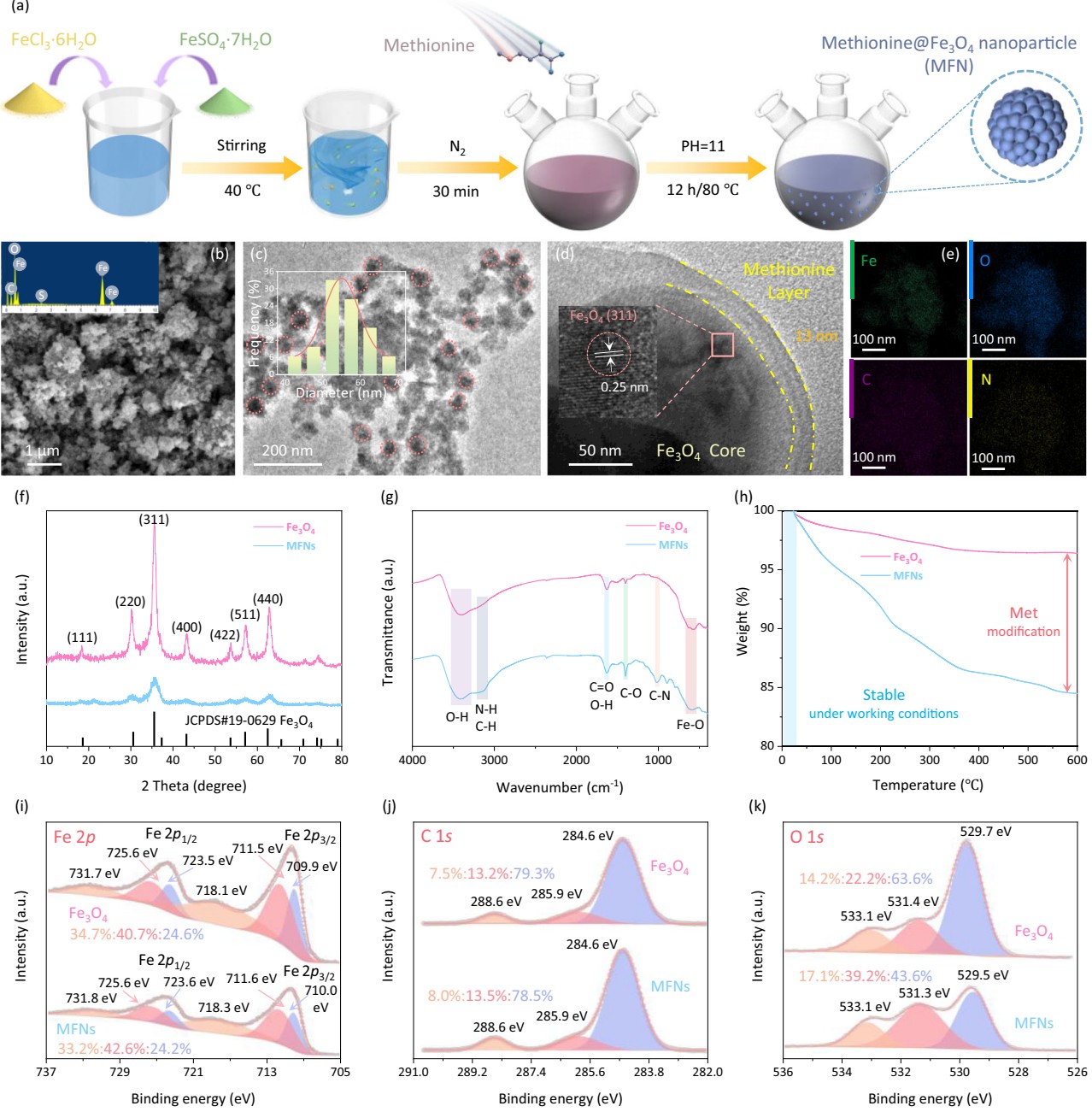

**Fig. 2 | Fabrication process, morphology, phase and structural characterizations of MFNs (Methionine@Fe₃O₄ nanoparticles). a** Schematic illustration for the synthesis of MFNs. **b** SEM image of MFNs. The inset in **b** shows the EDS analysis of MFNs. **c** Transmission electron microscopy (TEM) image of MFNs. The inset in **c** shows the size distribution of MFNs. **d** High-resolution transmission electron microscopy (HRTEM) image of MFNs. The inset in **d** shows the lattice fringes of MFNs. **e** Elemental mapping images of MFNs. **f** XRD patterns of $Fe_3O_4$ nanoparticles and MFNs. **g** FTIR spectra of $Fe_3O_4$ nanoparticles and MFNs. **h** TG curves of $Fe_3O_4$ nanoparticles and MFNs. The blue background in **h** indicates the temperature condition for application and storage of MFNs (-2–30 °C). **i–k** High-resolution XPS spectra of Fe 2p (**i**), C 1s (**j**), and O 1s (**k**) of $Fe_3O_4$ nanoparticles and MFNs. Source data are provided as a Source Data file.

methionine layer with a thickness of ~13 nm was distinctly wrapped on the surface of $Fe_3O_4$ core, the clear lattice fringes with a spacing of 0.25 nm correspond to (311) plane of $Fe_3O_4$. The energy-dispersive X-ray spectroscopy (EDS) spectra (inset of Fig. 2b) and the elemental mapping images as well as their overlaps (Fig. 2e and Supplementary Fig. 1a-c) revealed the homogeneous distribution of Fe, O, N, C, and S elements, further demonstrating the encapsulation of $Fe_3O_4$ by methionine substrate and the well-dispersion of $Fe_3O_4$ grains. The core-shell structure imparted magnetic properties to MFNs and protected the magnetic core from oxidation. Detailed values of the relative content of Fe, O, C, N, and S elements of MFNs obtained by EDS analysis (inset of Fig. 2b) are displayed in Supplementary Table 3.

Figure 2f displays the X-ray diffraction (XRD) patterns of MFNs and the counterpart of bare magnetic $Fe_3O_4$ powder. For MFNs sample, a series of peaks located at 18.3°, 30.2°, 35.6°, 37.1°, 43.0°, 53.3°, 57.1°, and 62.9° could be ascribed to the (111), (220), (311), (222), (400), (422), (511), and (440) planes of magnetite $Fe_3O_4$ (JCPDS no. 19-0629)[44,45], respectively, and the crystalline structure of $Fe_3O_4$ was well-retained. Compared with bare $Fe_3O_4$, a distinct decrease in the XRD peak intensities could be observed, indicating the prominent coating effect of methionine shell. In comparison with the XRD pattern of methionine solids (Supplementary Fig. 2), no diffraction peaks corresponding to methionine crystals were detected, indicating that methionine existed in an amorphous form on the surface of $Fe_3O_4$ rather than a distinct crystalline phase. In addition, a slight negative shift within 0.1° of diffraction peaks could be attributed to lattice distortions caused by macroscopic residual stresses[46]. Fourier transform infrared spectroscopy (FTIR) was conducted to investigate the chemical composition of MFNs and $Fe_3O_4$. As displayed in Fig. 2g, the peak located at 581 $cm^{-1}$ corresponded to the intrinsic stretching vibration of Fe–O bond of $Fe_3O_4$. The broad peaks in the range of 3700 to 3150 $cm^{-1}$ and at 1627 $cm^{-1}$ corresponded to the bending vibration of O–H bond. The spectra of MFNs showed an intense broadband at ~3420 $cm^{-1}$ due to the overlap of N–H, C–H and O–H stretching of the carboxylic group[47]. The characteristic peaks at 1630 $cm^{-1}$ and 1400 $cm^{-1}$ corresponded to C=O and C–O stretching vibrations of carboxyl of methionine, respectively, significant shifts of these two peaks compared to pure methionine indicate that methionine was coupled to $Fe_3O_4$ through its carboxyl group (Supplementary Fig. 3). In addition, the wavenumber interval between the symmetric and asymmetric stretching vibrations of $COO^-$ (230 cm) reflected their interaction corresponding to monodentate coordination[48]. FTIR spectra confirmed that methionine was successfully coated on $Fe_3O_4$ grains, and the absence of dense peak bands corresponding to pure methionine and significant peak shifts indicated a monolayer distribution rather than aggregates. As shown in Fig. 2i-k and Supplementary Fig. 4, the surface chemical states of MFNs and $Fe_3O_4$ were investigated by X-ray photoelectron spectroscopy (XPS). The XPS survey spectrum (Supplementary Fig. 4a) of MFNs clearly detected the signals of Fe, O, C, and S, which were consistent with EDS results. Figure 2i shows the Fe $2p$ signals with two separate peaks centered at 724.5 eV and 710.6 eV are attributed to Fe $2p_{1/2}$ and Fe $2p_{3/2}$ from $Fe_3O_4$ nanocores. Fe $2p$ can be divided into satellite peaks, $Fe^{3+}$ ($2p_{1/2}$, 725.6 eV; $2p_{3/2}$, 711.5–712.6 eV) and $Fe^{2+}$ ($2p_{1/2}$, 723.5–723.6 eV; $2p_{3/2}$, 709.9–710.0 eV). The peak area ratio of $Fe^{2+}$ to $Fe^{3+}$ for MFNs was almost similar to that of $Fe_3O_4$, indicating the synthesis of magnetite nanoparticles ($Fe_3O_4$). For the C $1s$ spectra in Fig. 2j, the peaks located at 284.6 eV were attributed to surface graphite-like carbon and used as a reference. The remaining two peaks of C $1s$ spectrum located at 285.9 eV and 288.6 eV revealed the existence of C–N/C–O/C–S, and C=O, respectively. As shown in Fig. 2k, the O $1s$ of XPS was divided into three categories, including lattice oxygen ($O_{latt}$, 529.5–529.7 eV), oxygen substances adsorbed on the surface ($O_{abs}$, 531.3–531.4 eV), and molecular $H_2O$ adsorbed on the surface ($O_{surf}$, 533.1 eV)[49,50]. The increase in the area of $O_{abs}$ peak

from 22.2% to 39.2% demonstrated the successful introduction of surface carboxyl, carbonyl, and hydroxyl oxygen species. Weak signal of S $2p$ spectrum was attributed to the introduction of methionine during polymerization (Supplementary Fig. 4b)[51]. XPS results verified the complexation of magnetic $Fe_3O_4$ cores and surficial methionine in the MFNs samples. The amount of methionine grafted onto MFN was determined from Thermogravimetric (TG) results (Fig. 2h), and the mass percentage of methionine in MFN was calculated to be ~11.91%. MFNs were structurally stable under applied temperature conditions (2–15 °C) and ambient storage environment (20–30 °C).

## CO$_2$ capture performance and mechanism of MFNs via CO$_2$@Water co-crystallization

For a better comparison of the role of MFNs and conventional promoters in $CO_2$@Water co-crystallization, pure water, $Fe_3O_4$ nanoparticles and SDS solution (with optimal concentration-0.05 wt%) were set as controls. Typical $CO_2$@Water co-crystallization characteristics, including nucleation and growth kinetics, were illustrated in Fig. 3. Working condition (275 K, 40 bar) was adopted to provide sufficient subcooling and adequate $CO_2$ supply to weaken crystallization randomness with a driving force of $\Delta P = \sim 2.2$ MPa for reliable performance comparison. Pressure-Temperature (P-T) evolution could be divided into four stages, including cooling and $CO_2$ dissolution, clathrate nucleation, crystal rapid growth and slow growth stage. The induction time (clathrate crystal nucleation stage) is defined as the time from the moment when P-T condition first reaches the phase equilibrium curve of $CO_2$ clathrate ($t_{eq}$) to the moment when the first clathrate crystal appears accompanied by a sudden increase in temperature ($t_f$, as shown in Fig. 3e). Subsequently, gaseous $CO_2$ molecules were trapped in hydrogen-bonded water cages and transformed into solid $CO_2$@Water clathrates, which grew downward from the gas-liquid interface and were accompanied by capillary-driven interfacial movement (Fig. 3f)[52]. The detailed formation process of hydrogen-bonded water cages was elucidated by molecular dynamics (MD) simulations (Fig. 3g and Supplementary Figs. 5 and 6). Details about MD simulations are shown in the Supplementary Note 1. Initially dispersed water molecules continuously rearranged around $CO_2$ via hydrogen bonds to form a locally ordered cage-like structure, which gradually evolved into a complete water cage with the addition of hydrogen bonds. Water cage stabilized with $CO_2$ trapping as the reduction in the free energy of the system. The formed water cage expanded via shared hydrogen bonds between water molecules to form a larger hydrogen bonding network to continuously trap $CO_2$ molecules. Stable water cages were represented as $5^{12}$ cage (regular pentagonal dodecahedron) and $6^2 5^{12}$ cage (tetrahedron consisting of two opposite hexagons and 12 pentagons arranged between them). Water cage exhibited some transient irregular shapes during $CO_2$ capture affected by the continuous breaking and rearrangement of hydrogen bonds, which were clearly displayed in the Supplementary Movies 1 and 2.

$CO_2$@Water clathrate failed to form in pure water during ~700 min. In contrast, both SDS and nanoparticles could accelerate the clathrate nucleation process to shorten the induction time (Fig. 4d). The induction time of $CO_2$@Water co-crystallization in $Fe_3O_4$-0.5 dispersion was reduced from 95.8 ± 38.7 min to 69.5 ± 48.0 min compared to SDS solution, which suggested that high conductive metal nanoparticles could enhance heat and mass transfer of solution to promote clathrate nucleation. Unfortunately, crystal growth was stagnant after a short period of rapid growth, and the final $CO_2$ capture capacity of 39.4 ± 0.5 v/v was lower than that of 79.0 ± 6.2 v/v for SDS (Fig. 4a and b), attributing to the strong surface tension-reducing capacity and additional clathrate nucleation sites provided by micellar particles of SDS[53]. As shown in Fig. 4d, with the introduction of methionine, induction time of crystallization was reduced by 37% in

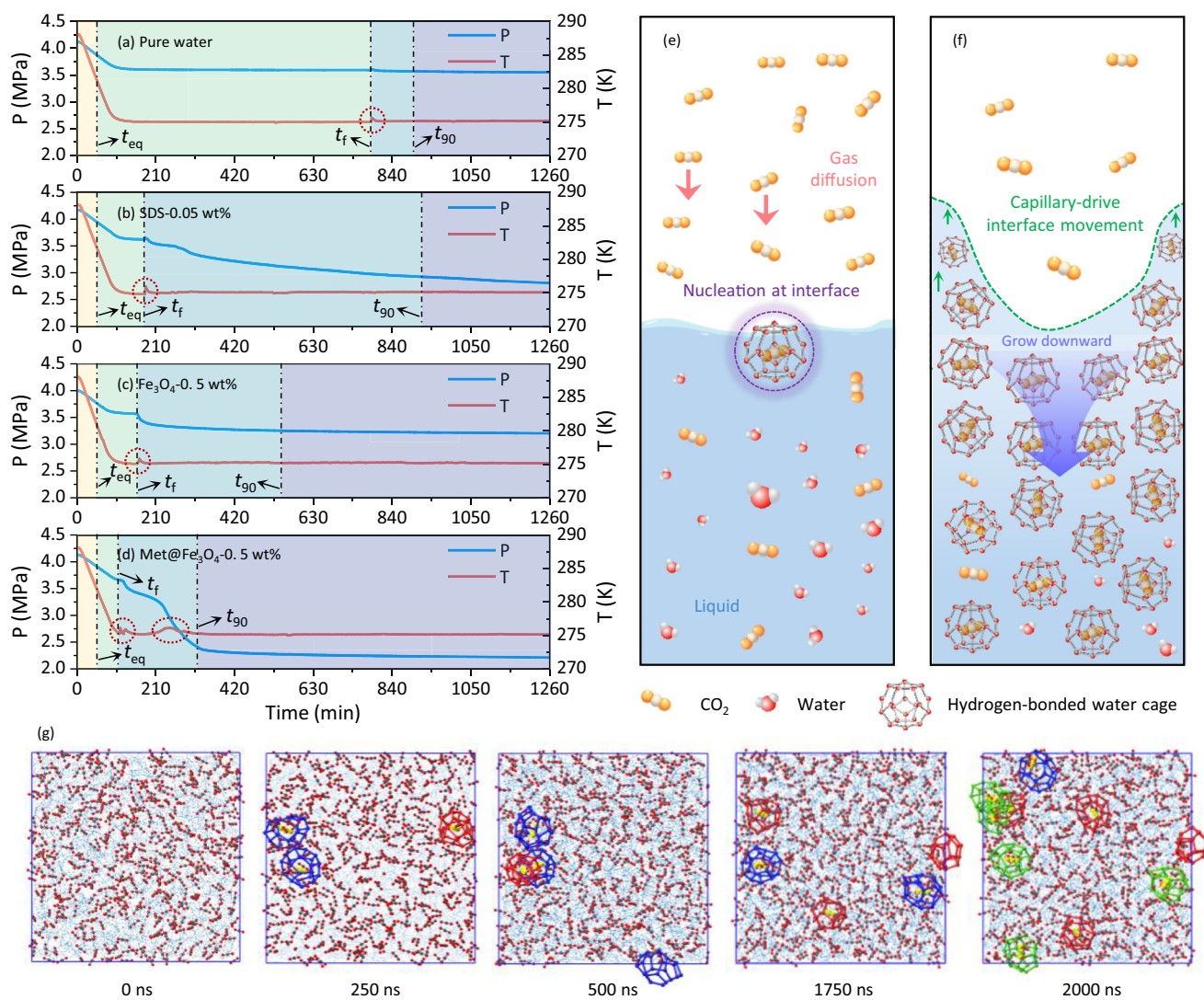

**Fig. 3 | Comparison of crystallization kinetics of CO₂@Water in different systems.** Pressure and temperature evolution during CO₂@Water clathrate formation in **a** pure water, **b** 0.05 wt% SDS solution (SDS-0.05), **c** 0.5 wt% Fe₃O₄ nanoparticles dispersion (Fe₃O₄-0.5) and **d** 0.5 wt% MFNs dispersion (MFNs-0.5), respectively. $t_{eq}$ refers to the moment when P-T condition first reaches the phase equilibrium curve of CO₂ clathrate, $t_f$ refers to the initial time of clathrate formation, and $t_{90}$ refers to the time required to achieve 90% CO₂ final gas uptake in clathrate. **e** and **f** depict schematic diagrams of CO₂@Water clathrate nucleation and growth, respectively.

**g** Snapshots of MD simulation at different moments for continuous CO₂ capture in hydrogen-bonded water cages. Free CO₂ molecule is represented as a small ball-and-stick model consisting of black and red colors, and captured CO₂ molecule is appropriately enlarged and changed to yellow for differentiation. Water molecules are represented as light blue lines. Red cage is $5^{12}$ water cage, blue cage is $6^2 5^{12}$ water cage, and green indicates incomplete or otherwise irregular cage structures. Source data are provided as a Source Data file.

MFNs dispersion compared to SDS control group. CO₂ capture capacity curves suggested that CO₂ capture in MFNs dispersion consistently took the lead from the beginning to the end of the process, both during clathrate nucleation and crystal growth stage (Fig. 4a). As displayed in Fig. 4b and c, ~80% water of MFNs-0.5 dispersion was involved in forming hydrogen-bonded water cages for CO₂ fixation, and the ultimate CO₂ capture capacity was as high as 117.5 ± 2.1 v/v; CO₂ capture volume was 6.8 and 1.5 times relative to pure water and SDS sample, respectively. As shown in Fig. 4e $t_{90}$ of CO₂ clathrate formation in MFNs-0.5 dispersion was shortened to 352.9 ± 33.9 min, which was only 43% of that in SDS solution (820.4 ± 98.3 min). Instantaneous growth rate after clathrate nucleation of SDS solution (1.450×10mol/min) was not much weaker than that of MFNs dispersion (1.620×10mol/min), and the fatal defect of capture capacity was caused by the stagnation of CO₂@Water clathrate growth in SDS solution after a short period of rapid growth (Fig. 4f). The subsequent slow kinetics of CO₂ capture in SDS solution could be attributed to the

mass transfer limitation due to initial rapid formation of CO₂@Water in localized region at the gas-liquid interface hindered subsequent gas-liquid contact[39,52]. Interestingly, this spell was broken in the MFNs dispersion by granting a sufficiently golden period for the secondary rapid growth of CO₂@Water (Fig. 4f). CO₂@Water co-crystallization consists of three steps: CO₂ enters the liquid phase from the gas-liquid interface (step I), CO₂ diffuses to the clathrate growth interface in liquid (step II), and CO₂ molecular are trapped (step III)[54]. Hence, a possible mechanism for the promotion of MFNs on CO₂@Water crystallization kinetics was proposed (Fig. 4h). MFNs created micro-convection in an aqueous solution owing to an irregular motion promoted the diffusion of CO₂ molecules in water[55]; dispersed MFNs adsorbed abundant CO₂ molecules at nano-interfaces for local enrichment to enhance heterogeneous nucleation process of CO₂@Water[56]; hydrophobic side chains of methionine anchored to the surface of MFNs could induce water molecules to form an ordered tetrahedral network through hydrogen bonding[57]. This was

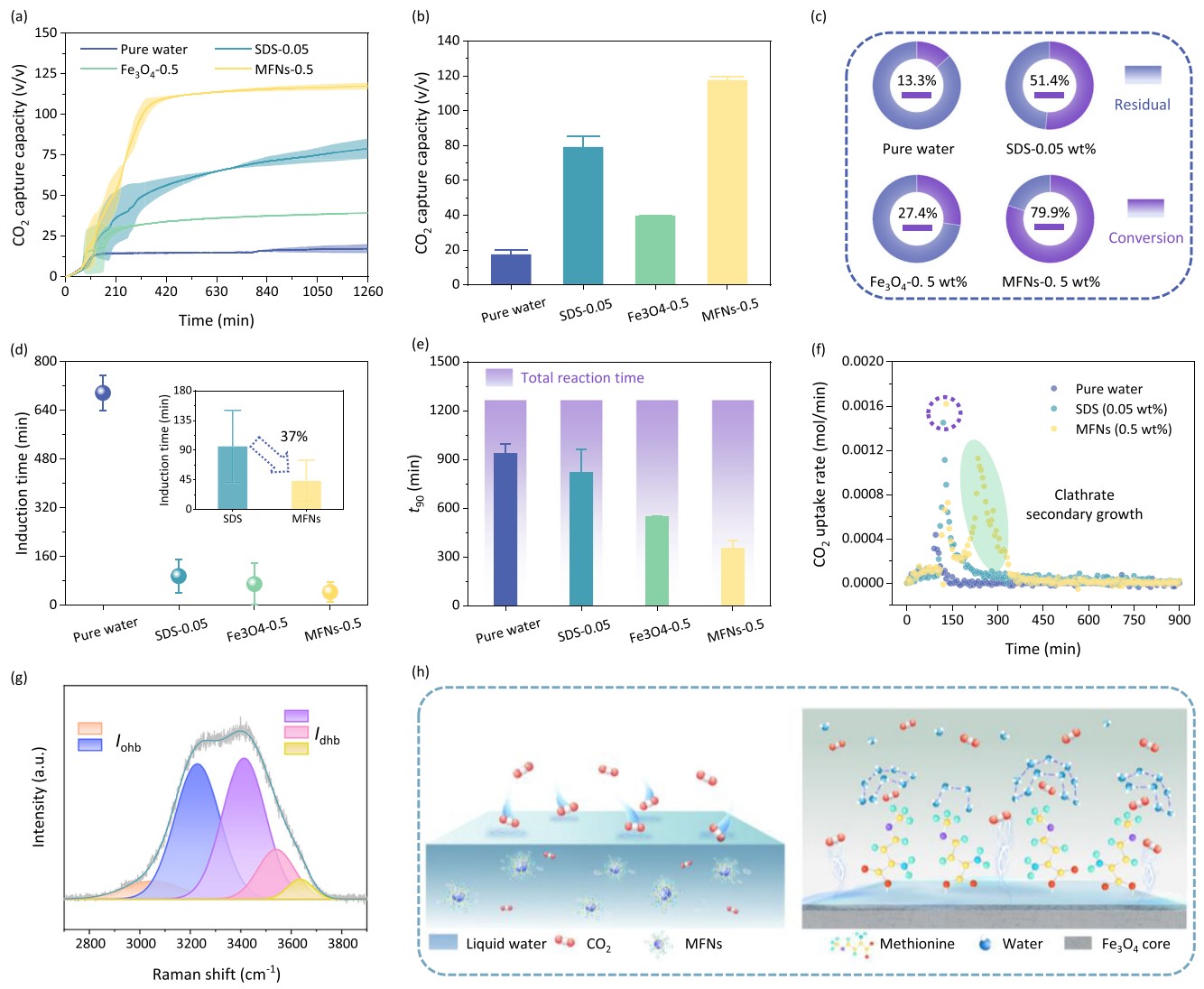

**Fig. 4 | $CO_2$ capture characteristics of MFNs-0.5, SDS-0.05, Fe₃O₄-0.5, and pure water via clathrate formation. a** Time-dependent $CO_2$ capture amount. **b** Ultimate $CO_2$ capture capacity. **c** Water conversion. **d** Induction time. The inset in **d** shows the comparison of induction time between MFNs-0.5 and SDS-0.05. **e** $t_{90}$ of $CO_2$ capture. Purple columns indicate the total reaction time. **f** Comparison of $CO_2$ uptake rate of MFNs-0.5, SDS-0.05, and pure water. **g** Raman spectrum of water stretching vibration in MFNs-0.5 dispersion and five Gaussian fitting peaks. **h** Mechanism of MFNs-enhanced $CO_2$ capture via $CO_2$@Water crystallization. For MFNs-0.5, Fe₃O₄-0.5 and pure water, the error bars in all these figures are determined from the standard deviation of four independent experiments; for SDS-0.05, the error bars in all these figures are determined from the standard deviation of five independent experiments. Source data are provided as a Source Data file.

corroborated by the ordering of hydrogen bonding arrangement of water molecules by quantifying Raman spectra of water stretching vibrations in MFNs-0.5 dispersion (Fig. 4g, a detailed Raman peak analysis was described in Supplementary Note 2). $I_{ohb}/I_{dhb}$ of MFNs-0.5 dispersion raised from 0.68 to 0.82 relative to pure water, implying an ordered tetrahedral network arrangement of water molecules in solution (Supplementary Fig. 7), laying the foundation for the subsequent formation and growth of water cages. In addition, the excellent thermal/mass transfer capability of fluid improved by MFNs provided assistance for $CO_2$ diffusion and co-crystallization of $CO_2$@Water. The thermal diffusion coefficient and thermal conductivity of the fluid were significantly enhanced by MFNs (Supplementary Fig. 8a and d), which contributed to the maintenance of subcooling and the release of latent heat of crystallization. As shown in Supplementary Fig. 8b and e, the promotion on the nucleation of $CO_2$@Water crystal was corroborated by in-situ differential scanning calorimetry (DSC), with the pronounced peaks of heat flow release in MFNs dispersion compared to pure water, indicating the drastic nucleation of hydrogen-bonded water cages. As shown in

Supplementary Fig. 8c and f, in-situ time-varying Raman spectroscopy during the $CO_2$ capture process showed that MFNs improved the growth stagnation (within 25–45 min) of hydrogen-bonded water cages in pure water, and the maintenance of excellent reaction kinetics was attributed to the improvement of heat-mass transfer conditions in fluid. Details of in-situ DSC and Raman experiments for $CO_2$ capture in hydrogen-bonded water cages were shown in the Supplementary Notes 3 and 4. These results showed that MFNs were preferable to provide an effective promotion effect for $CO_2$ capture via co-crystallization.

The dependence of $CO_2$@Water crystallization on the concentration of MFNs under the same operating condition was investigated to determine the generalizability of MFNs (Fig. 5). The ultimate $CO_2$ uptake in the pure water (without MFNs) was generally low (<20 v/v). The formation of a monolayer $CO_2$@Water clathrate film at the gas-liquid interface prevented the interaction between $CO_2$ and liquid water. As MFNs concentration increased from 0.05 to 0.2 wt%, ultimate $CO_2$ capture capacity gradually increased from $42.9 \pm 13.3$ v/v to $118.7 \pm 2.3$ v/v, reflecting the positive correlation effect of MFNs

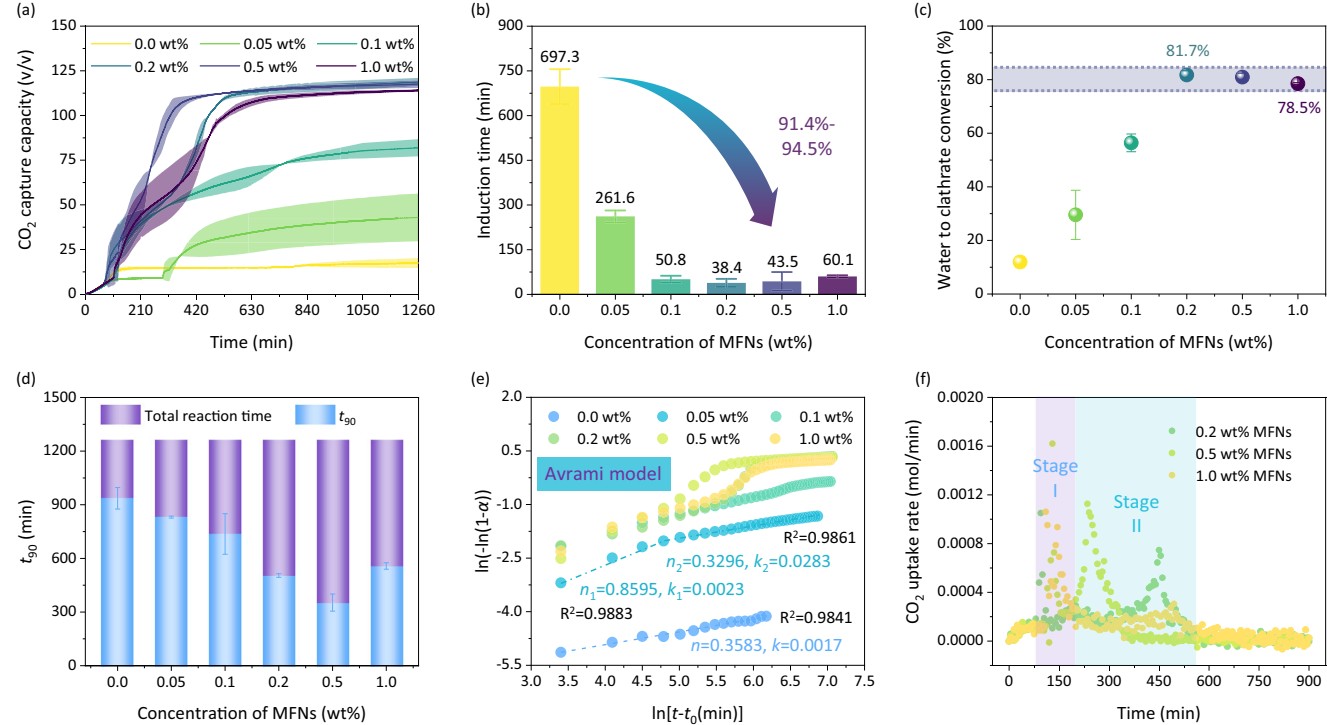

**Fig. 5 | CO₂ capture characteristics of MFNs dispersion (0.05-1.0 wt%). a** Time-dependent CO₂ capture capacity. **b** Induction time. **c** Water to clathrate conversion. The narrow grey shaded region in **c** indicates similar water to clathrate conversion ratios obtained by 0.2–1.0 wt% MFNs dispersions. **d** $t_{90}$. **e** Kinetic model of CO₂ capture via crystallization. $\alpha$ represents the water-to-clathrate conversion ratio, $t$ represents the reaction moment, and $t_0$ refers to the time of clathrate nucleation. **f** Comparison of CO₂ uptake rate of MFNs-0.2, MFNs-0.5 and MFNs-1.0. For MFNs-0.05, the error bars in all these figures are determined from the standard deviation of eleven independent experiments; for other systems, the error bars in all these figures are determined from the standard deviation of four independent experiments. Source data are provided as a Source Data file.

concentration on CO₂@Water co-crystallization. However, further raising MFNs concentration to 1.0 wt% slightly impaired the promotion of MFNs on CO₂@Water clathrate formation, resulting in CO₂ capture capacity maintaining in the range of 114.0–118.7 v/v and the water conversion maintaining in the range of 78.5%–81.7% (Fig. 5c). The presence of MFNs which provided crystal nucleation sites could effectively reduce energy barrier and shorten induction time of CO₂@Water clathrate nucleation (Fig. 5b). There still remained a longer induction period (261.6 ± 14.1 min) of clathrate nucleation in MFNs-0.05, and the nucleation barrier persisted until MFNs concentration was elevated to 0.1 wt%. Induction times of CO₂ clathrate nucleation ranged from 38.1 to 60.1 min within 0.1 to 1.0 wt% MFNs dispersion, whose values were shortened by 91.4–95.3% compared with pure water case (697.3 ± 41.3 min). However, while MFNs-0.1 significantly accelerated clathrate nucleation, the improvement in subsequent crystal growth was limited, with an ultimate CO₂ capture capacity of 81.9 ± 4.8 v/v. There were heat-mass transfer limitations owing to the irregular motion of MFNs was not intense enough and the small size effect of MFNs was less influential in nanoparticle dispersion with low concentration[54]. The time-dependent CO₂ capture capacity curves in Fig. 5a indicate that CO₂@Water crystal growth kinetics exhibit a significant dependence on MFNs concentration.

Avrami phase transition model (the detailed description was shown in Supplementary Note 5) was used to describe CO₂@Water crystallization kinetics (Fig. 5e), the fitness of the model was demonstrated in Supplementary Fig. 9a and d[58]. The value of Avrami exponent $n$ in the initial stage varied from 0.3583 to 1.0161 (Supplementary Figs. 9 and 10), which suggested diffusion-controlled one-dimensional growth of CO₂@Water crystal in pure water or MFNs dispersions. Non-integer values of $n$ illustrated that CO₂@Water nucleation occurred anywhere from instantaneously to sporadically[59]. The lower value of $n$

in pure water was attributed to the mass transfer resistance of CO₂ diffusion through the liquid film. Increases in MFNs concentration yielded a raise in the number of stages in CO₂@Water clathrate formation and evolution curve. The formation of a clathrate film after rapid CO₂@Water crystallization in MFNs-0.05/0.1 limited further mass transfer, which was reflected in the decrease in $n$ value (Supplementary Fig. 9b and c). For MFNs dispersions with high concentrations (0.2, 0.5 and 1.0 wt%), the sharp increase of $n$ in stage II/III was attributed to multiple nucleation of CO₂@Water, specifically the crystal growth leaded to fluctuation in the nearby composition, which increased nucleation rate in region of neighboring growing nuclei, triggering a chain-reaction-like process (Supplementary Fig. 10)[60,61]. The final growth stage was the slowest where crystallization process almost ceased due to extensive clathrate growth in the interspaces, thus limiting further contact of liquid water and CO₂. CO₂@Water crystal growth kinetics showed an elevated tendency with increasing concentration of MFNs, as reflected in $t_{90}$ (Fig. 5d). A comparison of CO₂@Water clathrate formation kinetics in MFNs-0.2, 0.5, and 1.0 with similar final CO₂ capture capacity was emphasized. As MFNs concentration increased from 0.2 wt% to 0.5 wt%, $t_{90}$ was shortened from 504.3 ± 7.3 min to 352.9 ± 33.9 min, achieving a prominent kinetic facilitation in CO₂@Water formation. The increase in nucleation sites, the more ordered arrangement of water molecules, and the improved thermal conductivity of aqueous solution endowed a high-quality heat-mass transfer environment for CO₂@Water crystallization in MFNs-0.5 dispersion. However, $t_{90}$ of MFNs-1.0 increased by 58.0% compared to MFNs-0.5 (Fig. 5d), and ultimate CO₂ capture capacity was slightly decreased (-3.5 v/v). For MFNs dispersions with high concentrations, Brownian motion of nanoparticles was limited by the increase of viscosity[62], which leaded to a reduction in mass transfer coefficient; moreover, the surface effect and small size effect of nanoparticles

would be weakened by the agglomeration of magnetic MFNs. The agglomeration of nanoparticles was reflected in the hydrodynamic diameter evolution with increasing concentration (Supplementary Fig. 11b-f), especially MFNs showed significant agglomeration in the 1.0 wt% system. The agglomeration was consolidated by the decreasing absolute value of zeta potential with increasing concentration (Supplementary Fig. 11a), which implied a gradual weakening of MFNs dispersion. In addition, due to the hydrophilic property of MFNs (Supplementary Fig. 12), the −OH groups on MFNs surface could form hydrogen bonds with water molecules, competing for the clathrate cage construction and inhibiting $CO_2$@Water crystallization. For MFNs dispersions with high concentrations, the inhibitory effects were pronounced, which weakened crystallization kinetics and ultimate $CO_2$ capture capacity. Figure 5f clearly shows the two-stage growth characteristics of $CO_2$@Water crystal in MFNs-0.2, 0.5, and 1.0. $CO_2$ clathrate growth rates were almost identical during the short period after nucleation (Stage I), and the differences in crystallization kinetics occurred mainly in the secondary growth period (Stage II). Compared to MFNs-0.5, the secondary growth of $CO_2$ clathrate was significantly weak in MFNs-0.2 and MFNs-1.0, and there was a distinct time lag between primary and secondary clathrate growth. $CO_2$ capture performance was affected by mutual constraints of concentration and dispersibility of MFNs in solution. Hence, MFNs-0.5 was identified as the optimal candidate for crystallization-based $CO_2$ capture by comparing indicators including $CO_2$ capture capacity, reaction rate and duration.

## Recyclability and biocompatibility analysis of MFNs for application potential assessment

$Fe_3O_4$ cores endow MFNs with magnetic recycling and reusing properties, which will reduce material cost and avoid environmental pollution. As shown in Supplementary Fig. 13a, the saturation magnetization of MFNs (25.5 emu/g) was lower than that of bare $Fe_3O_4$ (54.3 emu/g) owing to the addition of methionine-shell reduced the specific gravity of $Fe_3O_4$ in MFNs; but MFNs still provided enough response sensitivity to varying magnetic fields. The coercivity ($Hc$) and remanent magnetization ($Br$) values of the samples were below 30 Oe and 0.2 emu/g, respectively. Such superparamagnetism is essential for MFNs application in solution, preventing self-magnetic aggregation for effective recycling[63]. As displayed in Supplementary Fig. 7b, MFNs could be efficiently separated from water within 1 min by a simple external magnetic field. Effective separation of MFNs and clean water realizes resource recovery and avoids water waste and environmental pollution caused by irreversible separation of traditional water-soluble clathrate promoters and water. $CO_2$@Water promotion performance of separating and reusing MFNs-0.5 with a magnet for 17 cycles are displayed in Fig. 6, which were conducted in three parallel sets. $CO_2$ capture capacity profiles of the 17 cycles behaved similarly and no significant mass transfer limitations were observed during reaction time (Fig. 6a). $CO_2$ capture capacity was maintained above 109.37 v/v over 17 cycles, which demonstrated the excellent reusability of MFNs (Fig. 6b). The promotion effect of MFNs on clathrate formation kinetics was slightly weakened with the raise of cycle number (reflected in the increase of $t_{90}$), suggesting a potential change in material property during the recycling test. XRD patterns of fresh and recycled MFNs were consistent (Supplementary Fig. 14a), indicating the crystalline phase structure of $Fe_3O_4$ remained stable and amorphous methionine did not crystallize. The comparison of FTIR spectra (Supplementary Fig. 14b) showed that the characteristic peak of C=O, C−O, and O−H derived methionine remain consistent, implying that there was no degradation and decomposition of methionine. As shown in Supplementary Fig. 14d-f, Fe $2p$ and C $1s$ signals of MFNs before and after recycling test remained consistent; the $O_{latt}$ ratio in the O$1s$ spectra of recycled MFNs was raised compared to that of fresh MFNs, and the slight attenuation of $O_{abs}$ and $O_{surf}$ signals suggested a small

amount of desorption of methionine. TG curves indicated that the weight loss of MFNs during recycling test was 2.84%, which quantified the methionine desorption (Supplementary Fig. 14c). Continuous $CO_2$ capture process based on hydrogen-bonded water cages involves a variety of physical and chemical interactions, including mechanical disturbances, hydrodynamic forces, local stress changes associated to crystallization, and $CO_2$ migration during decomposition. Mechanical effects such as friction with the reactor, shear forces from stirring-enhanced fluid motion, collisions between nanoparticles, and extrusion from the expansion of water crystallization would inevitably deplete the methionine loads. Same reactor, consistent operating conditions, and similar crystalline capacity (relative error within 6.1%), implying that the materials were subjected to almost identical mechanical effects during recycling tests. TG curves of two MFNs tested over 17 cycles (Supplementary Fig. 15) showed consistent weight loss trends, with the final difference being within 0.5%, implying that the MFNs failed in the same pattern. Ultimate $CO_2$ capture capacity showed MFNs maintained excellent promotion performance with continuous recycling, but the reaction kinetics gradually deteriorated. The failure time of MFNs was defined from reaction kinetics, where the average $t_{90}$ exceeding 90% of the reaction time was considered as material failure. In this study, $t_{90}$ of $CO_2$ capture reached 631 min after 17 cycles, exceeding 90% of the total reaction time (700 min), thus determining that the material failed during this cycle, with the failure time being the sum of the reaction times of first 16 cycles (11200 min). The kinetics of $CO_2$@Water formation in the MFNs dispersion was still reassuring compared to SDS, with $t_{90}$ remaining below 12.4% of that in SDS solution, which was obtained by the Avrami model (Supplementary Fig. 16).

Foam generation during crystal decomposition directly hampers the economics of crystallization promoters[64]. Foam generated on the surface of SDS solution is inclined to be blown out by gas flow during clathrate decomposition, thus impairing reusability; additional separation and purification processes to remove SDS mixed into the gas stream add to the cost burden. As shown in Supplementary Fig. 17, the foam in SDS solution was generated in large quantities during $CO_2$@Water decomposition process and was hard to self-eliminate. In contrast, foams were almost nonexistent in MFNs dispersions, on the one hand, tiny $CO_2$ bubbles burst instantly once transferred to the gas-liquid interface affected by high surface tension; on the other hand, Brownian motion of nanoparticles suspended in solution during clathrate decomposition also contributed to the rupture of generated bubbles[65,66]. Overall, MFNs show great potential for future applications for the economic benefits brought by foam inhibiting properties, simple and efficient recovery characteristics, and recycling performance. According to the comparison of costs and $CO_2$ capture capacity between MFNs and SDS (Supplementary Table 4), although MFNs are not cost-competitive initially, their excellent recyclability and promoting performance make them advantageous for long-term and continuous $CO_2$ capture process. This indicates that MFNs with high-performance stability, are suitable for long-term sustainable $CO_2$ capture, with their economic advantages becoming more pronounced over extended recycling periods.

Toxicity assessment of $CO_2$@Water clathrate promoter is a crucial indicator for their large-scale application for the protection of natural environment and human health. Real-time cell analysis (RTCA) was performed to evaluate the cytotoxicity of the as-prepared MFNs and SDS (the most typical representative of traditional clathrate promoters) against Chinese hamster ovary (CHO) cells. CHO-K1 cells used in our study originated from the subcloning of CHO cells in deep biopsies of adult Chinese hamster ovaries, which was a widely used model for providing an indication of the potential toxicity in vivo[67,68]. The optimal number of 5000 inoculated cells was chosen for cytotoxicity assessment determined by monitoring proliferation curves of CHO cells with gradient initial numbers (Supplementary Fig. 18).

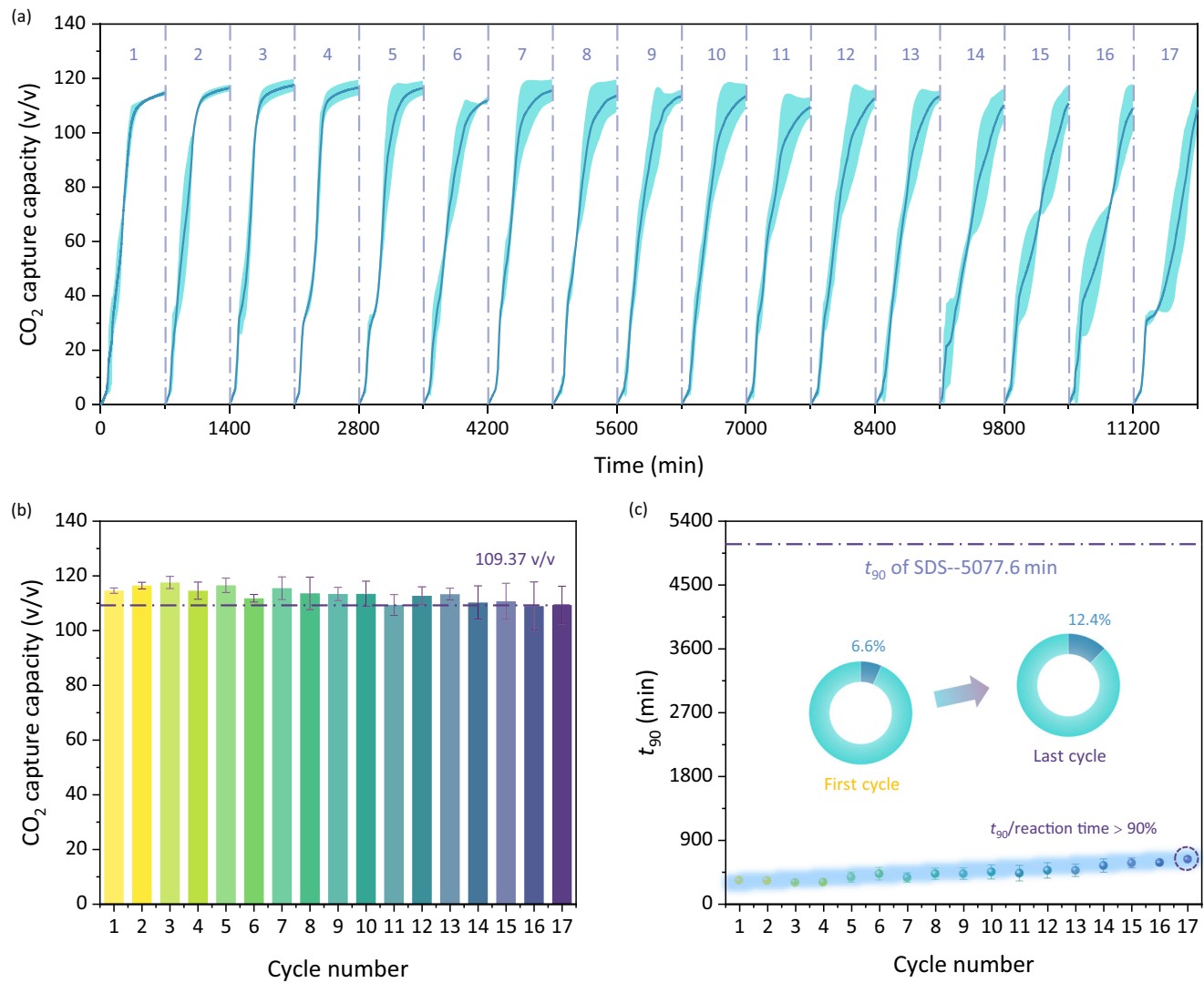

**Fig. 6 | CO₂ capture performance of MFNs-0.5 in 17 individual recycling experiments. a** Time-dependent $CO_2$ capture curves. **b** Ultimate $CO_2$ capture capacity. **c** $t_{90}$. The two inserts in **c** show $t_{90}$ of the first cycle and the last cycle in MFNs-0.5 as a proportion of that in SDS solution, respectively. The blue background line behind the data points in **c** indicates that $t_{90}$ shows an increasing trend as the cycle number increases. The error bars in all these figures are determined from the standard deviation of three independent measurements. Source data are provided as a Source Data file.

Primarily, cells were inoculated into three electronic microtiter plates (E-Plates) and propagated freely for ~25 h; then different concentrations of SDS and MFNs dispersions were introduced into the plates, and cell growth was recorded for 50 h with a blank control (Fig. 7a). Cells exposed to MFNs exhibited a similar proliferation trend to control check (CK), although their growth was slightly inhibited after ~10 h (Fig. 7b). The normalized cell index did not change significantly within 20 h, but then MFNs exhibited a slight concentration-dependent inhibition on cell proliferation. Unfortunately, the introduction of SDS was almost disastrous for cell growth. Figure 7b shows that a mere 50 mg/L of SDS resulted in cell proliferation arrest, and the cells underwent massive apoptosis with SDS concentration increased to 200 mg/L. Figure 7c provides a more visual comparison of the time-dependent cell viability of MFNs and SDS. Cell viability in the presence of SDS showed a regular sharp decrease trend with exposure time, it was reasonable to hypothesize that cell viability will tend to 0 as the indefinite extension of exposure time. Besides, cell viability was greatly affected by the concentration of SDS, which remained above 78% for the first 12 h at low concentration (50 mg/L), but decreased to 60% in just 3 h at high concentration (200 mg/L). Cell viability in MFNs showed a decreasing and then increasing trend, which might be related to cellular uptake and excretion. As shown in Fig. 7d, cell viability was still higher than 81% incubated with 200 mg/L MFNs for 48 h, contrarily more than 95% of cells died during incubation after introduction with 200 mg/L SDS for 48 h. The results showed that MFNs had significantly lower cytotoxicity and good biocompatibility compared with SDS.

Low regeneration energy consumption and high fixation capacity are key prerequisites to the widespread popularization and application of $CO_2$ capture technology. As shown in Fig. 8a, energy consumption of $CO_2$@Water phase transition can be as low as 57.1 kJ/mol $CO_2$, which is significantly lower than the enthalpy changes of most $CO_2$ absorbents and approaches to the physisorptive domain (detailed information is displayed in Supplementary Table 5). This avoids the energy-intensive processes of conventional absorbents recovery or crystalline precursors regeneration. Moreover, the stubborn mass transfer barrier of $CO_2$@Water crystallization was eliminated by MFNs. Compared with the previously reported liquid/solid inducers, MFNs exhibited an obviously improved "catalytic activity" for $CO_2$@Water crystallization (Fig. 8b and Supplementary Table 6), and their $CO_2$ trapping performance was almost close to that of state-of-the-art porous carbon and MXene (Fig. 8c and Supplementary Table 7). Hence, inducing water

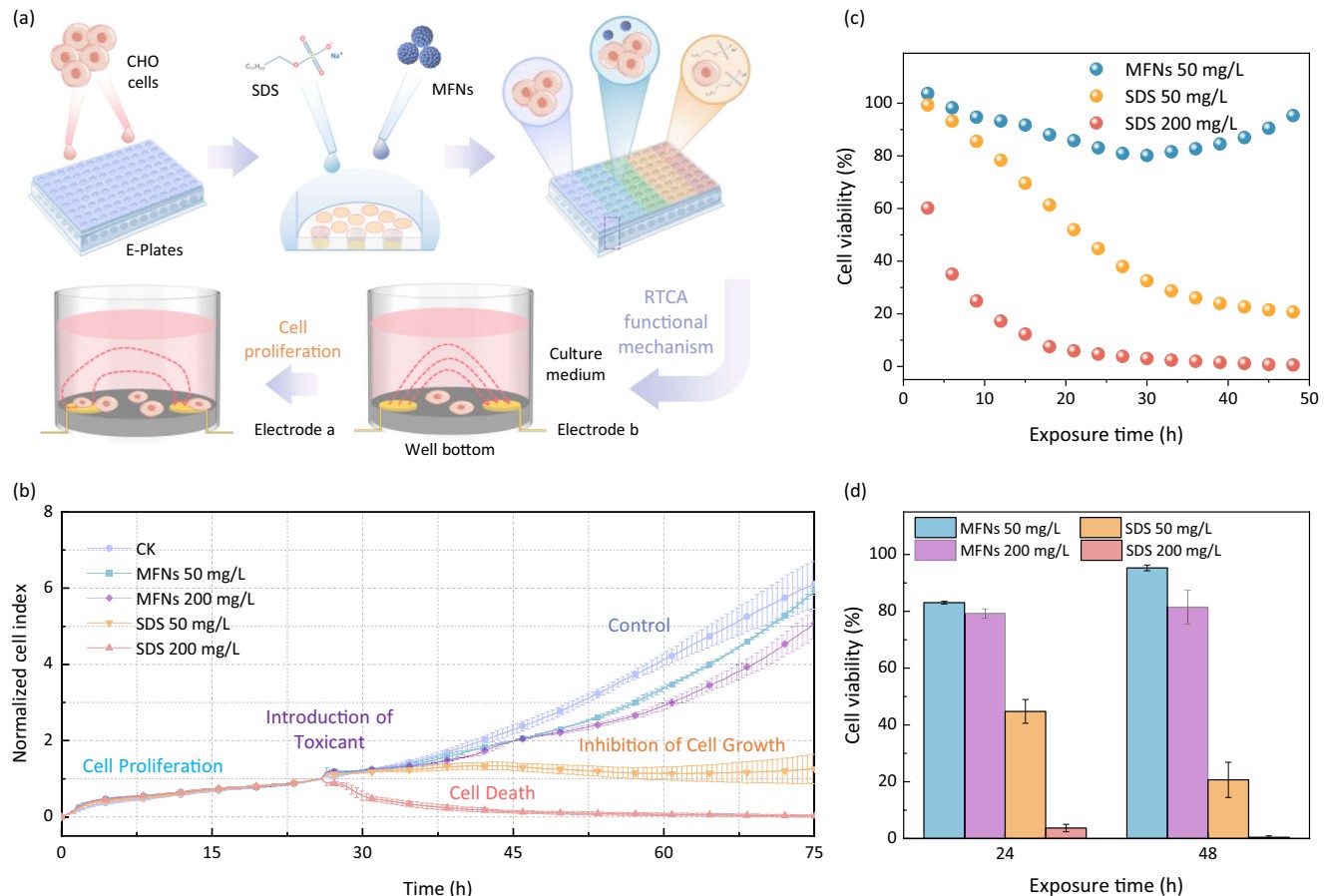

**Fig. 7 | Cytotoxicity studies of CO₂@Water clathrate promoters including MFNs and SDS. a** RTCA system and procedure for cytotoxicity assessment. **b** Cell proliferation profiles incubated with MFNs/SDS (50/200 mg/L). **c** 48 h sustained monitor of cell viability influenced by SDS (50/200 mg/L) and MFNs (50 mg/L).

**d** Cell viability incubated with MFNs/SDS for 24 and 48 h at different concentrations. The error bars in all these figures are determined from the standard deviation of three independent measurements. Source data are provided as a Source Data file.

molecules to construct hydrogen-bonded nanocages for CO₂ fixation via reversible phase transitions is highly attractive for sustainable carbon capture and storage. For the practicality and industrial feasibility, cooling energy demands can be reduced by leveraging industrial waste heat recovery or ambient cooling in appropriate settings. Some gas guests (such as ethane, propane and isobutane) from industrial sources can be used as additives to increase the operating temperature by weakening crystallization energy barrier to alleviate energy demand[69]. Moreover, the regeneration of MFNs is achieved through a simple room-temperature heat exchange, eliminating the requirement for energy-intensive desorption processes. These features highlight the feasibility and sustainability of the proposed method, particularly for industrial applications where low-grade heat sources are available.

## Discussion

In summary, herein we propose a crystallization-based ZRD method for efficient CO₂ capture via hydrogen-bonded water cages induced by organo-magnetic nanoparticles with core-shell structure. Methionine was anchored to the surface of magnetic Fe₃O₄ cores, the rearrangement of water molecules induced by organic promoter and the small size effect of inorganic nanoparticles synergistically improved the mass-transfer conditions for CO₂@Water co-crystallization. CO₂ was trapped in water cages induced by MFNs via pressure swing crystallization with a high capacity (118.7 v/v) and density (22.7 wt%). MFNs with an optimal concentration shortened the nucleation of CO₂@Water to less than 38.4 min and enhanced CO₂ gas uptake by 5.8 and 0.5 times compared with pure water and SDS solution. 17

independent cycle tests totaling 11900 min demonstrated the recycling performance of MFNs, and the foam generation issue of conventional clathrate promoters during CO₂ release was effectively avoid. Magnetism endowed MFNs effective separation properties from water, realizing the recycling of metal and clean water resources and avoiding environmental problems. More importantly, the desirable biocompatibility of MFNs demonstrated by cytotoxicity evaluation provided a safe guarantee for their application and promotion. We hope this work would provide an energy-efficient and environmentally friendly strategy for future sustainable CO₂ capture and storage.

## Methods

### Materials

Reagents including ferric chloride (AR, 99.5%, FeCl₃·6H₂O), ferrous sulfate (AR, 99.5%, FeSO₄·7H₂O), sodium hydroxide (AR, 96%, NaOH), L-Methionine (99%, C₅H₁₁NO₂S) and anhydrous ethanol (AR, 99.5%) were obtained and used without any further purification from Macklin Biochemical Co., Ltd, Shanghai. Deionized water (18.2 MΩ·cm) was supplied by a Milli-Q system. CHO-K1 cells were obtained from Procell Life & Technology Co., Ltd., and cultured in dulbecco's modified eagle medium supplemented with 1% penicillin/streptomycin and 10% fetal bovine serum with 5% CO₂ at 37 °C.

### Synthesis of MFNs

MFNs were synthesized by the controlled co-precipitation method. Briefly, 20 mmol of FeCl₃·6H₂O and 10 mmol of FeSO₄·7H₂O were dissolved in 100 mL deionized water at 40 °C. Then, 20 mL of 2 mol/L

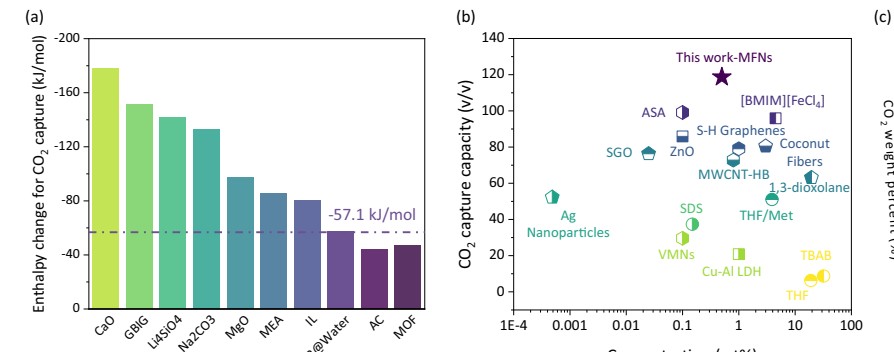

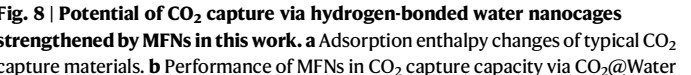

**Fig. 8 | Potential of CO₂ capture via hydrogen-bonded water nanocages strengthened by MFNs in this work. a** Adsorption enthalpy changes of typical CO₂ capture materials. **b** Performance of MFNs in CO₂ capture capacity via CO₂@Water crystallization compared with reported promoters. **c** A comparison of CO₂ capture capacity of MFNs-induced CO₂@Water crystals and typical porous materials. Source data are provided as a Source Data file.

L-methionine solution was added to the mixture in a nitrogen atmosphere and after 30 min, aqueous NaOH solution (2 M) was added to the reaction mixture until pH value raised to 11.0 ± 0.2. After addition, the mixture was stirred at 80 °C and 300 rpm for 12 h. Finally, MFN precipitates were separated from the solution by centrifugation at 4000 rpm and washed with deionized water and anhydrous ethanol to remove possible residues and then vacuum-dried at room temperature.

## Characterizations

FE-SEM images and EDS were obtained by a NOVA NanoSEM scanning electron microscope with the electron beam voltage at 5 kV and 10.0 μA. TEM images were obtained using a JEM-1400Flash at 120 kV, and corresponding elemental mappings were collected when required. The enlarged STEM image was denoised by low-pass filtering. Hydrodynamic diameter distribution of MFNs was obtained using a Malvern Zetasizer Nano ZS90 at 298 K. XRD was conducted on a Bruker D8 Advance X-ray diffractometer with Cu Kα radiation (λ = 0.15406 nm, operated at 40 kV and 40 mA) in the range of $2\theta = 10–80°$. FTIR spectroscopy was investigated by Thermo Scientific Nicolet 6700 with a 400–4000 cm$^{-1}$ range. XPS was conducted using a Thermo Scientific ESCALAB 250Xi X-ray photoemission spectrometer with a region from 0 to 800 eV. The scan chamber was pumped to a high vacuum (10Pa) before data acquisition. X-ray energy was set to 15 keV with the Fe (2p), O (1s), C (1s), and S (2p) elements targeted for high-resolution scans. Investigative scan with a large energy range were performed on each sample prior to high-resolution scans on individual elements. The magnetic property was measured by hysteresis loops from 0 to ± 20,000 Oe on a physical property measurement system equipped with a vibrating sample magnetometer (PPMS-VSM, Quantum Design, Dynacool) at 298 K. Thermogravimetric analysis (TGA, Q-500) was conducted in nitrogen atmosphere from room temperature (25 °C) up to 600 °C at a heating rate of 10 °C/min. The thermal diffusion coefficient and thermal conductivity of pure water and 0.5 wt% MFNs dispersion at ambient temperature (298 K) and experimental temperature (275 K) were measured using a Differential Scanning Calorimeter (DSC 300, NETZSCH, Germany) and a Thermal Conductivity Analyzer (LFA467, NETZSCH, Germany).

## CO₂ capture procedure based on hydrogen-bonded water cages

MFNs were dispersed ultrasonically in 20 mL deionized water for 30 min, and then the solution was placed in a pressure-resistant stainless-steel reactor. The reactor was flushed three times with CO₂ at 0.5 MPa to completely remove air, and the final pressure in the reactor was fixed at 40 bar with a mechanical stirring at 400 rpm for 30 min to achieve CO₂ dissolution equilibrium. The reactor was cooled from 288 K to 275 K at a rate of 0.15 K/min by an external circulating glycol bath

with continuous stirring at 100 rpm. Nucleation of CO₂@Water crystals was identified by the detected temperature rise. After the induction period of nucleation, CO₂ was continuously captured by hydrogen-bonded water cages and formed crystal precipitation. The reaction time was controlled to be 1260 min for the performance tests of different materials and MFNs concentration, and 700 min for the independent experiments in the recycling performance tests. The experiments for performance tests were repeated under controlled conditions based on extensive pre-tests, and the number of independent trails depended on the reproducibility varying from at least 4 to more than 10. CO₂ capture capacity was calculated to be −v/v by Volume Balance-Mass Conservation (VB-MC) shown in Supplementary Note 6. The standard deviations of capture capacity and crystallization kinetic indicators were represented in the form of error bars in the figure.

The effects of classical gas clathrate crystallization promotor SDS and bare Fe₃O₄ nanoparticles on CO₂@Water clathrate formation were investigated. Crystallization-based CO₂ capture was processed using a similar procedure, except that different additive was added to 20 mL deionized water.

## Recovery of MFNs

The reactor containing CO₂@Water clathrate crystals was transferred to ambient temperature, and MFNs dispersion regeneration was achieved as crystals decomposition by simple heat exchange. CO₂ detrapping process was monitored by pressure evolution, with a constant pressure indicating that all CO₂ escaped from hydrogen-bonded water cages. MFNs were separated from the suspension by an external magnetic field applied by a magnet and then recovered by vacuum drying.

## Cytotoxicity assessment

RTCA technique was used to determine the cytotoxicity of MFNs and SDS on CHO cells. RTCA system could convert interfacial impedance signal into a "cell index" via a microelectronic biosensor to monitor cell proliferation in real-time, label-free, and non-invasively. Cell index is a dimensionless parameter, calculated by the electrical impedance signal from E-plate, which can reflect the cell physiological and functional state. Graded numbers of CHO cells were inoculated on E-plates to determine the optimal initial number of inoculated cells to achieve a cell index of 1 within ~25 h propagation. MFNs/SDS dispersed in deionized water were diluted with culture fluid before adding to E-plates, and the blank controls (cell culture media) were conducted concurrently. Once the cell index reached 1, MFNs dispersion or SDS solution with different concentrations were added to the cultures, and subsequent cytotoxicity testing continued for 50 h. Cell index value was automatically updated every 15 min until the end of experiment. Cell indices in all plate wells were normalized to effectively compare

the effect of toxicants on cell viability. Cell viability was expressed as percentage of normalized cell index (NCI) of the test group to control group, and the NCI of the blank control group was set as 1. Triplicate tests were conducted for each sample.

## Data availability
The authors declare that the data supporting the findings of this study are available in the main article, Supplementary Information and Source data file. Source data are provided with this paper.

## Code availability
All molecular dynamics simulations were performed using the open-source GROMACS package, which is available at https://www.gromacs.org; The VMD software used for structural visualizations is available at https://www.ks.uiuc.edu/Research/vmd.

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

## Acknowledgements

This work was supported by the National Natural Science Foundation of China (Grant No. 52006024).

## Author contributions

T.W., L.Z., A.H. and Y.S. conceived the idea and designed the project. L.Z. and Y.S. supervised the research. T.W. performed the experiments and analyzed the data. M.F. carried out the cytotoxicity assessment. T.W. wrote the original draft. C.M. offered insightful suggestions on manuscript preparation and edited the original draft. L.Z. and A.H. revised the original draft. All authors discussed the results and commented on the paper.

## Competing interests

The authors declare no competing interests.
