## [Transparent Peer Review file · Nature Communications]

Organic magnetic nanoparticles catalyze CO₂ capture in hydrogen-bonded nanocages via water-driven crystallization

Corresponding Author: Dr Aliakbar Hassanpouryouzband

Version 0:

Reviewer comments:

Reviewer #1

(Remarks to the Author)

Wang et al. present a manuscript on the synthesis of Methionine@Fe₃O₄ nanoparticles (MFNs) for CO₂ fixation through hydrogen-bonded nanocages. The topic is relevant and addresses an important global challenge in CO₂ capture. However, while the manuscript presents some interesting findings, there are several areas that require significant improvement for it to meet the publication standards. Below are detailed comments:

1. Introduction and Significance:

The introduction outlines the general approach but could benefit from a clearer and more focused presentation of the study's objectives and the specific contributions of this work compared to existing CO₂ capture technologies. Highlighting the novelty and relevance of the proposed method would help set the stage for the reader more effectively.

The claim that the preparation of porous carbons is "lengthy and cumbersome" could be better supported with more concrete examples or references, as this may not apply universally to all cases.

2. CO₂ Adsorption Mechanism:

The manuscript suggests that CO₂ is captured within hydrogen-bonded water cages, but the details surrounding the formation and function of these cages are unclear. Providing more robust characterizations and a clearer explanation of this mechanism would greatly strengthen the key claims of the paper. This aspect is critical and should be elaborated further.

The CO₂ storage capacity reported (118.7 v/v, 22.7 wt%) is promising, but it would be useful to offer a more comprehensive comparison with current state-of-the-art materials to put the results into context. This would help clarify where this method stands in terms of performance.

3. Experimental Conditions and Methodology:

The CO₂ sorption experiments were conducted at 275 K and 40 bar, which are relatively energy-intensive conditions. It would be beneficial to explain the rationale behind choosing these conditions and whether this process could be optimized for real-world applications. Addressing the energy efficiency and scalability of the method is important for practical implementation.

Some crucial experimental details are missing, such as specific conditions during the electrochemical CO₂ capture, temperature, and pressure. Providing these details is important for reproducibility.

4. Material Characterization:

More information is needed on the Methionine layer (Met). Is it crystalline or amorphous, and what is its thickness? These details are important to understand how the material interacts with Fe₃O₄ and affects the CO₂ adsorption process.

The characterization of the mono-layer structure in Figure 4h is not sufficiently supported by the TEM images alone.

Additional characterization methods should be considered to verify this structure.

5. Reusability and Stability:

The manuscript reports on the stability and reusability of the MFNs, but this section would benefit from more comprehensive data on the long-term performance of the material. Including insights into potential degradation mechanisms or factors that could affect long-term stability would provide more depth to this analysis.

While the results are promising, a more thorough comparison with alternative materials, both in terms of performance and cost, would give readers a better understanding of the potential impact of this method.

6. Literature Review:

The literature review could be strengthened by including more recent advancements in CO₂ capture technologies.

Comparing this electrochemical approach with other emerging methods, such as those using porous carbons or zeolites, would provide a more comprehensive view of the field.

7. Figures and Tables:

The figures are informative but could benefit from more descriptive captions. Including key experimental details or takeaways in the captions would improve clarity.

A comparative table summarizing the performance of various CO₂ capture technologies discussed in the manuscript would be a useful addition.

8. References:

There are some formatting inconsistencies in the references that should be addressed. Ensuring uniformity in the reference style would enhance the professionalism of the manuscript.

Conclusion:

While this manuscript presents an interesting approach to CO₂ capture, there are several significant areas that need improvement, particularly in terms of clarifying the CO₂ adsorption mechanism, enhancing material characterization, and providing more detailed experimental data. Additionally, a more comprehensive comparison with current CO₂ capture technologies would help to contextualize the findings.

Reviewer #2

(Remarks to the Author)

The manuscript presents an innovative and potentially impactful approach to carbon capture using a crystallization-based method facilitated by Methionine@Fe₃O₄ nanoparticles. This strategy enhances the CO₂ capture process by inducing hydrogen-bonded water nanocages, which demonstrates significant improvements over traditional crystallization-based CO₂ capture methods. The use of magnetic nanoparticles with biocompatibility and recyclability characteristics positions this work as a novel contribution to the field of sustainable CO₂ capture technology. However, while the research is of high quality and offers substantial promise, a few minor revisions are necessary before publication.

1. Some terms in the supplementary equations lack unit definitions. Appropriate units should be included for all variables.
2. The statement in the manuscript, line 140, regarding the EDS elemental mapping images as well as their overlaps (Fig. 2e and Supplementary Fig. 1) revealed the homogeneous distribution of Fe, O, N, C, and S elements..." is not fully supported by the data presented. The core-shell structure and particle size (~75 nm) of MFNs are not clearly visible in Figure 2e. Higher-resolution imaging is needed for validation.
3. In Figure 3, which shows the "Pressure and temperature evolution during CO₂@Water clathrate formation,". The X-axis of Figure 3 should include "Time (min)" to ensure proper interpretation of the figure.
4. For recycling performance of the MFNs-0.5, were the results shown in Figure 6 reproducible? Did each sample fail at the same point and in the same way? How long did it take for the average nanoparticles to fail? Provide details on the number of trials and statistical analysis for the recycling performance in Figure 6 to confirm the robustness of the results.
5. The authors suggest that the proposed method holds great potential for industrial-scale CO₂ capture, but there is little discussion on how this would be achieved practically. For example: The claim of phase transition energy for CO₂ capture using MFNs is as low as 0.57 kJ/mol CO₂, which is indeed lower than other methods. However, a detailed energy balance for the entire capture process, including cooling, crystallization, and regeneration, is required to evaluate the industrial feasibility of this method.
6. The manuscript mentions that increasing MFNs concentration beyond 0.5 wt% led to reduced performance due to agglomeration and viscosity effects. However, this is only briefly discussed and needs a more in-depth exploration. If agglomeration of nanoparticles is a limiting factor in this system, it could hinder its scalability. Have you considered methods to prevent nanoparticle agglomeration, and what are the potential trade-offs between preventing agglomeration and maintaining CO₂ capture efficiency?
7. The explanation of how Methionine@Fe₃O₄ nanoparticles induce local water ordering and enhance CO₂ capture kinetics needs more detailed mechanistic insights. Although the manuscript provides some discussion around hydrogen bonding and micro-convection, further experimental evidence (such as fluid dynamics data) or theoretical modeling would help to support these claims.

Reviewer #3

(Remarks to the Author)

In this paper, Wang et al. presented organic magnetic nanoparticles, (Methionine@Fe₃O₄, termed as MFNs), which trapped CO₂ via pressure swing crystallization with a high capacity (118.7 v/v) and density (22.7 wt%), thus providing a new inspiration for sustainable CO₂ capture with zero resource depletion (ZRD). In addition, MFNs have good biocompatibility, which provides a safety guarantee for their application and promotion. The specific design concept of this work is novel and unique. The manuscript provides valuable insights into the development of future sustainable CO₂ capture and storage. I agree that this manuscript to be accepted after some minor revision. The following are the suggestions for the authors to improve the manuscript.

1. The aggregation problem of magnetic nanoparticles can lead to a decline in material performance and affect the dispersibility in water. Discuss whether it has a certain impact on CO₂ capture performance.
2. XRD, XPS, and EDS etc. abbreviations should be expanded in full for the first time they appear in the text and then used in abbreviated form thereafter. For example, XPS is expanded in line 173 and EDS in line 503, rather than being expanded for the first time they appear.
3. References should be added to supported the conclusions of lines 101-103 in manuscript.
4. The picture definition is low. Please check the full text.

Version 1:

Reviewer comments:

Reviewer #1

(Remarks to the Author)

The authors have addressed all my concerns, and the paper can be accepted with present form.

Reviewer #2

(Remarks to the Author)

The authors did a very careful revision. I can accept their arguments and recommend ACCEPT.

Reviewer #3

(Remarks to the Author)

The authors carefully revised the manuscript to address any concerns and improve the overall quality of the paper. It is now acceptable for publication.

Response to reviewers

General response:

- *Thank you for your detailed and constructive feedback on our manuscript. We appreciate the opportunity to revise our work based on the comments from the reviewers and the editorial team. Your insights have been very helpful in improving the focus and clarity of the article, and we are grateful for your guidance.*

Point-to-point response

Reviewer #1 (Remarks to the Author):

Wang et al. present a manuscript on the synthesis of Methionine@Fe₃O₄ nanoparticles (MFNs) for CO₂ fixation through hydrogen-bonded nanocages. The topic is relevant and addresses an important global challenge in CO₂ capture. However, while the manuscript presents some interesting findings, there are several areas that require significant improvement for it to meet the publication standards. Below are detailed comments:

- *Thank you for all your insightful comments. We have followed your suggestions to improve the quality of the article.*

1. Introduction and Significance:

The introduction outlines the general approach but could benefit from a clearer and more focused presentation of the study's objectives and the specific contributions of this work compared to existing CO₂ capture technologies. Highlighting the novelty and relevance of the proposed method would help set the stage for the reader more effectively. The claim that the preparation of porous carbons is "lengthy and cumbersome" could be better supported with more concrete examples or references, as this may not apply universally to all cases.

- *Many thanks for your valuable comments, which are essential to the quality improvement of this work.*
This study aims to propose an innovative strategy for sustainable CO₂ capture that combines high storage density, environmental friendliness and energy efficiency. We further emphasize this research objective clearly (Page 6 in the revised manuscript): "Herein, a crystallization-based ZRD scheme combined high storage density, environmental friendliness and energy efficiency is formulated for sustainable CO₂ capture via reversible phase transition". Conventional CO₂ capture technologies are dominated by liquid amine absorption, physical adsorption by porous materials, and carbonate crystallization. Liquid amine absorption is the most mature method to capture CO₂, but it has a series of defects, including high regeneration temperature, equipment corrosion, environmental pollution and complicated wastewater treatment caused by degradation products (such as ammonia, sulfate, nitrate, etc.). CO₂ capture by porous material adsorption is an emerging and efficient technology, but there are some challenges in practical applications: high-performance materials such as zeolites, metal-organic frameworks (MOFs), covalent organic frameworks (COFs), etc., usually require high

preparation costs and complicated synthesis procedures; material poisoning due to moisture or impurity adsorption can significantly impair CO₂ capture efficiency; moreover, although its regeneration energy consumption is typically lower than that of liquid amine absorption, heat or vacuum conditions are still required for CO₂ desorption. The advantages of carbonate crystallization for CO₂ capture are mild reaction conditions and controllable products, while the drawbacks are high energy consumption required for carbonate regeneration, significant water demand for carbonate dissolution and reaction, and the scaling due to bicarbonate or carbonate crystallization and precipitation. In comparison, CO₂ capture within hydrogen-bonded water cages in liquid water possesses commendable unique advantages that enable sustainable and green CO₂ capture, i.e.,: (a) low energy consumption for regeneration (under ambient temperature and pressure conditions), (b) the primary raw material is merely natural liquid water (no toxic chemicals), which makes the reaction process environmentally friendly, (c) the captured CO₂ exists in the form of solid crystals, which is less prone to leakage and facilitates and convenient for storage and transportation, and (d) insensitivity to industrial gas impurities (such as N₂, O₂, NO, H₂O, etc.), resulting in a low risk of equipment corrosion.

Although the technology of capturing CO₂ within hydrogen-bonded water cages has shown prominent potential for sustainability and energy conservation, a fatal flaw of slow natural crystallization kinetics severely limits its industrial application. While some surfactants and nanomaterials with high specific surface areas have been found to improve the heat and mass transfer conditions in the liquid phase to facilitate crystallization, these chemical promoters are costly, challenging to recycle and reuse, and prone to residue in liquid water resulting in water wastage and potential environmental pollution. The green crystallization process involved in CO₂ capture based on hydrogen-bonded water cages points the way to sustainable CO₂ capture, and proposing environmentally friendly, durable, robust, and resource-saving schemes to enhance crystallization kinetics is the key to popularize the application. Therefore, in this study, high-performance organic magnetic nanoparticles (methionine@Fe₃O₄ nanoparticles-MFNs) have been designed to strengthen CO₂@Water crystallization, addressing the challenge of slow crystal nucleation. The properties of excellent durability, recyclability, and non-toxicity of this material endow it the potential to achieve zero resource depletion. The coupling of the unique properties of this high-performance material, which allow it to excel in crystallization-based CO₂ capture reactions, makes the research objective feasible. Benefiting from these unique properties, the coupling of the material and crystallization-based CO₂ capture reaction makes the research objectives feasible. In the revised manuscript, the research objective is further clarified and the novelty of the proposed methodology and its relevance to the research objective are emphasized.

Zeolites, porous carbon, metal-organic frameworks (MOFs), covalent triazine frameworks (CTFs), and covalent organic frameworks (COFs) are porous materials that excel in the field of CO₂ capture due to their high specific surface areas, abundant pore structures (tunable microporous, mesoporous, and macroporous structures), high adsorption selectivity, as well as excellent chemical and thermal stability. However, the preparation of high-performance porous materials requires a series of processes, including precursor selection, optimization of reaction conditions, and post-processing. MOFs synthesis involves the precise proportioning of metal source and organic ligand, the control of solvothermal reaction conditions, subsequent solvent replacement and activation. Porous carbon with high specific surface area, excellent chemical stability, and flexibly tunable pore structure, stands out as an excellent porous material for CO₂ adsorption. The preparation of porous carbon involves the selection of precursors, template guidance, carbonization, template removal, and modification. In the revised manuscript, specific literatures have been cited to illustrate the requirements of reaction conditions and

processes for the preparation of high-performance porous materials: " However, fine material design, multiple processes and stringent reaction conditions are necessary for the preparation of high-performance porous materials, such as template guidance, charring and template removal usually required for porous carbon with large pore volumes, and days-long crystal growth required for some MOFs preparation.^{18, 19"}

2. CO₂ Adsorption Mechanism:

The manuscript suggests that CO₂ is captured within hydrogen-bonded water cages, but the details surrounding the formation and function of these cages are unclear. Providing more robust characterizations and a clearer explanation of this mechanism would greatly strengthen the key claims of the paper. This aspect is critical and should be elaborated further. The CO₂ storage capacity reported (118.7 v/v, 22.7 wt%) is promising, but it would be useful to offer a more comprehensive comparison with current state-of-the-art materials to put the results into context. This would help clarify where this method stands in terms of performance.

- *Thank you very much for your kind reminder, and a detailed elaboration of hydrogen-bonded water cage formation is crucial for clearly elucidating the CO₂ capture mechanism. Under low-temperature and high-pressure conditions, dissolved CO₂ molecules in liquid water interact with water molecules through van der Waals forces and hydrogen bonds, gradually forming locally ordered structures within the liquid. As CO₂ molecules act as guests entering the initially formed cage-like structures composed of water molecules connected by hydrogen bonds, the free energy decreases and drives the stabilization of the cage structure to form nucleation sites. Once the initial hydrogen-bonded water cage is formed, surrounding water molecules further connect to the existing structure via hydrogen bonds, expanding into a larger and more complete hydrogen-bonded network. The continuous trapping of CO₂ molecules by water cages promotes the sustained growth and crystallization of the hydrogen-bonded water cages, culminating in the formation of macroscopic CO₂@Water solid crystals.*

As an important means for nanoscale molecular-level mechanism analysis, molecular dynamics (MD) simulations have been employed to reveal the mechanisms of CO₂-water interactions, water molecules rearrangement, and the step-by-step formation of hydrogen-bonded water cages. MD calculations were performed by Gromacs 2024.3, and visual analysis was conducted with VMD 1.9.3. The TIP4P/ice model was used for water molecules, and the EPM2 force field was applied to CO₂ molecules. The total simulation time was 2 μs, with a time step of 2 fs. A Nose-Hoover thermostat was used to regulate the temperature to 260 K, and a Parrinello-Rahman barostat was employed to control the pressure to 50 MPa. The temperature and pressure conditions used for the simulation were different from the experimental conditions because the actual nucleation and growth of macroscopic CO₂@Water crystals typically require several hours to even days, and co-crystallization of CO₂ and water molecules is nearly impossible in such conditions under the influence of high Gibbs free-energy barrier between solid and liquid phases. Therefore, ultra-low temperature and ultra-high pressure conditions were adopted to accelerate the nucleation of CO₂@Water crystals to elaborate on the characteristics and formation mechanism of hydrogen-bonded water cages. Prior to MD simulation, energy minimization was performed to eliminate unreasonable atomic contacts. Long-range electrostatic interactions with a cutoff radius of 1 nm were determined using the Particle Mesh Ewald (PME) method. Periodic boundary conditions were applied in all directions. The model contained 2944 water molecules and 512 CO₂ molecules, with an initial model size of 4.7 nm x 4.7 nm x 4.7 nm.

Figure 1.1 illustrates the simulation snapshots of the entire system at different moments within 2000 ns, depicting the formation and dynamic evolution of hydrogen-bonded water cages in detail. Figure 1.2 shows the evolution of the water cage structure within 0.5 nm around a CO₂ molecule involved in the nucleation of CO₂@Water crystal, transitioning from an irregular incomplete structure to a regular water cage. In the initial state, CO₂ molecules are randomly distributed in water, with water molecules predominantly in a disordered arrangement. Subsequently, CO₂ molecules weakly interact with neighboring water molecules through van der Waals forces, promoting the rearrangement of hydrogen bonds between the water molecules. Water molecules begin to spontaneously form pentagonal or hexagonal ring structures around the CO₂ molecules, which are the basic units of water cages. These localized water cages composed of ring-like structures are gradually refined into complete hydrogen-bonded water cages through continuous rearrangement and addition of hydrogen bonds. CO₂ molecules trapped in these water cages interact with the inner walls of cages through van der Waals forces, which reduces the free energy and further stabilizing the hydrogen-bonded water cages, implying the nucleation of CO₂@Water crystals. The cage structures expand through shared hydrogen bonds between water molecules, forming a larger hydrogen-bonded network. CO₂ molecules are continuously captured into unfilled water cages, promoting the continued growth of CO₂@Water crystals. The stable CO₂@Water crystals primarily contain two main types of cage structures: the 5¹² cage (regular pentagonal dodecahedron) and the 6²5¹² cage (tetrahedron consisting of two opposite hexagons and 12 pentagons arranged between them). In addition, the hydrogen-bonded network of water molecules during crystal growth is highly dynamic, with continuous breaking and reorganization of hydrogen bonds between water molecules. This non-static property allows the formed cage structures to potentially manifest as transient irregular shapes. Two visualizable videos have been added to the supplementary material, which more clearly depict the phenomenon of hydrogen bond breaking and re-forming between water molecules during the formation of hydrogen-bonded water cages. In addition, the formation of hydrogen-bonded water cages during CO₂ trapping process was confirmed by in-situ Raman spectroscopy.

Figure 1.1 Snapshots of the simulation at different moments in the system. Free CO₂ molecules are represented as a small ball-and-stick model consisting of black and red colors, and captured CO₂ molecules are appropriately enlarged and changed to yellow for differentiation. Water molecules are represented as light blue lines. Red cage is 5¹² water cage, blue cage is 6²5¹² water cage, and green indicates incomplete or otherwise irregular cage structures.

Figure 1.2 Structure of hydrogen-bonded water cages within 0.5 nm surrounding a CO₂ molecule involved in CO₂@Water crystal nucleation at different simulation moments.

Porous materials with high specific surface areas and significant adsorption capacities, such as metal-organic frameworks (MOFs), covalent organic frameworks (COFs), zeolites, and activated carbon, have been widely reported as state-of-the-art CO₂ capture materials. Consequently, CO₂ capture capacities of the current state-of-the-art materials were summarized in the CO₂ capture performance figure, which together with the previously selected representative advanced porous materials provide a comprehensive comparison of CO₂ capture capacities, which is essential to clarify the advantages of the present materials and methods in terms of performance. In addition, related temperature and pressure conditions employed in the reported studies have also been summarized in tables within the Supplementary Information to facilitate a more comprehensive comparison.

3. Experimental Conditions and Methodology: The CO₂ sorption experiments were conducted at 275 K and 40 bar, which are relatively energy-intensive conditions. It would be beneficial to explain the rationale behind choosing these conditions and whether this process could be optimized for real-world applications. Addressing the energy efficiency and scalability of the method is important for practical implementation. Some crucial experimental details are missing, such as specific conditions during the electrochemical CO₂ capture, temperature, and pressure. Providing these details is important for reproducibility.

- *Many thanks for your kind reminder to add the experimental details. In the revised manuscript (Pages 43-44), some main parameters of the experiments have been added in detail, including initial/final temperature-pressure conditions, cooling rate, stirring rate of the solution, total reaction time, and crystal dissociation procedure.*

CO₂@Water crystallization mainly depends on the stability of hydrogen bonds between water molecules and the trapping of CO₂ guest molecules. The formation of hydrogen-bonded network requires a stable thermodynamic environment, as strong thermal motion of water molecules at room or high temperatures can disrupt the orderliness of the hydrogen bonding lattice; low-temperature conditions reduce the kinetic energy of water molecules, thereby facilitating the arrangement of water molecules into regular cage-like structures through hydrogen bonds. Additionally, high-pressure conditions can significantly increase the solubility and density of CO₂ molecules, overcoming the tension in the hydrogen-bonded lattice between water molecules and promoting the capture of CO₂ guests by hydrogen-bonded water cages. Therefore, the formation and stability of hydrogen bonds are enhanced under low-temperature and high-pressure conditions, while the embedding of CO₂ molecules in water cages can further reduce free energy of the system, inducing the nucleation and growth of CO₂@Water crystals. In this study, 275 K and 40 bar were chosen as the experimental conditions for CO₂ trapping with the main purpose of providing sufficient subcooling and driving force to ensure CO₂@Water crystals could occur smoothly and avoiding the excessive randomness in crystallization caused by the insufficient thermodynamic conditions, which would affect the reliability of experimental results. Furthermore, this condition can effectively guarantee the crystallization kinetics to reduce time cost and ensure sufficient CO₂ supply to optimally compare the CO₂ capture capacity obtained with different materials.

For sustainable green CO₂ capture, appropriate pressure reduction or temperature enhancement is favorable for energy saving. Some larger gas guest molecules, such as ethane, propane, and isobutane, hold great potential for moderating the working conditions of hydrogen-bonded water cage-based CO₂ capture, which is conducive to ensure CO₂ capture capacity and reaction kinetics while reducing energy consumption. In the case of propane, (a) its hydrophobic effect in water induces the regular arrangement of water molecules, thus

promoting the formation of hydrogen-bonded cage structures and lowering the energy barrier required for CO₂@water crystallization; (b) CO₂ and propane molecules have complementary size characteristics, with a tendency to occupy small/large water cages, respectively, forming a more stable crystalline structure; (c) the preferential filling of large water cages by propane molecules contributes to moderate the free energy for CO₂@Water crystallization. Moreover, the CO₂-propane mixed gas has numerous industrial sources, including byproducts from refining industry, industrial flue gas treatment, and industrial syngas byproducts, which provide a flexible technical path for practical application. Two tests were conducted using 10% propane and 90% CO₂ mixed gas to replace pure CO₂ gas under the conditions of 275 K, 3 MPa, and 281 K, 4 MPa, respectively, achieving CO₂ capture capacities of 70.0 and 72.1 v/v in the 0.5 wt% MFNs dispersion, which is close to the promotional performance of surfactant SDS under 275 K and 4 MPa. Figure 1.3 shows the temperature and pressure evolution under the two conditions, with sharp temperature increases within 100 min after cooling indicating the nucleation of hydrogen-bonded water cages, and rapid pressure drops reflecting excellent crystal growth kinetics. This implies that the addition of 10% propane to CO₂ feed gas increased the experimental temperature by 6 K, reduced the operating pressure by 25%, while maintaining excellent reaction kinetics. Such mild operating conditions resulted in significant energy savings and facilitated the practical application of the green technology of CO₂ capture based on liquid water crystallization.

Figure 1.3 Pressure and temperature evolution during CO₂ capture under (a) 90% CO₂+10% propane, 3 MPa, 275 K and (a) 90% CO₂+10% propane, 4 MPa, 281 K.

4. Material Characterization: More information is needed on the Methionine layer (Met). Is it crystalline or amorphous, and what is its thickness? These details are important to understand how the material interacts with Fe₃O₄ and affects the CO₂ adsorption process. The characterization of the mono-layer structure in Figure 4h is not sufficiently supported by the TEM images alone. Additional characterization methods should be considered to verify this structure.

- Many thanks for this valuable suggestion, and detailed information about the methionine layer is crucial for elucidating the material characteristics. X-ray diffraction (XRD) analysis of methionine (Met) solid was performed using a Bruker D8 Advance X-ray diffractometer equipped with Cu K α radiation ($\lambda = 0.15406$ nm). The operating voltage was set at 40 kV, and the operating current was 40 mA, with a scanning range of $2\theta = 10-80^\circ$ and a step size of 0.02° .

The XRD pattern of pure Met solid is shown in Figure 1.4a, displaying distinct characteristic diffraction peaks (with main peak positions at 11.6°, 23.2°, 29.1°, 35.0°, 41.1°, and 47.3°), indicating a highly crystalline state. As shown in Figure 1.4b, the XRD pattern of MFNs did not show the diffraction peaks of methionine crystals at the corresponding positions, indicating that Met existed in an amorphous form on the surface of Fe_3O_4 rather than a distinct crystalline phase. Transmission electron microscopy (TEM) was used to obtain the micro morphology of MFNs, as shown in Figure 1.5. A clear loading of the Met organic layer on the surface of Fe_3O_4 can be observed, displaying distinct boundary striations, and the thickness analysis shows that the thickness of the Met layer is between 12.6 nm and 13.2 nm. This indicates that Met in MFNs is coated on the surface of Fe_3O_4 nanoparticles in the form of an amorphous layer.

Figure 1.4 (a) XRD pattern of pure Met. The inset shows a schematic and macroscopic view of Met molecular structure. (b) XRD patterns of Fe_3O_4 , MFNs, and Met. The signal value of Met is scaled down by a factor of 20 to highlight the remaining weak diffraction peaks.

Figure 1.5 HRTEM image of MFNs.

To elucidate the structure of MFNs, the FTIR spectrum of pure Met was obtained and compared with that of MFNs, as shown in Figure 1.6, to illustrate the interaction of Met with Fe_3O_4 . Two peak bands appearing at 1409 cm^{-1} and 1583 cm^{-1} are attributed to the symmetric (ν_{sym}) and asymmetric (ν_{asym}) stretching vibrations of COO^- group in Met, respectively. Compared to pure Met, these bands in MFNs are shifted to lower and higher wavenumbers (1400 cm^{-1} and 1630 cm^{-1}), indicating that Met is coupled to Fe_3O_4 through its carboxyl group. In addition, the difference in wavenumbers can reflect the interaction between Fe_3O_4 and Met. The wavenumber interval $\Delta\nu$ between ν_{sym} and ν_{asym} can be used to distinguish the type of

bonding between the carboxylate head and Fe atoms. A large $\Delta\nu$ (200-320 cm^{-1}) corresponds to monodentate coordination, a moderate $\Delta\nu$ (140-190 cm^{-1}) indicates bridging bidentate coordination, and a small $\Delta\nu$ (<110 cm^{-1}) corresponds to chelating bidentate coordination. For MFNs, $\Delta\nu$ (230 cm^{-1}) falls within the range of 200-320 cm^{-1} , suggesting that the interaction between carboxyl group and Fe_3O_4 is monodentate. In addition, the FTIR spectrum of MFNs did not exhibit dense peak bands corresponding to pure Met, indicating that Met did not form multilayer aggregates on the material surface. The distinct shift in the typical peak bands (such as the asymmetric stretching vibration of the COO^- group) of FTIR spectrum of MFNs compared to pure Met suggests that Met interacts with Fe_3O_4 in the form of chemical coordination and forms a monolayer distribution on the Fe_3O_4 surface. In the revised manuscript (Page 10), the coordination form of Met with Fe_3O_4 has been supplemented to fully elucidate the structure of MFNs, and the FTIR spectrum of Met has been added to the Supplementary Information (Supplementary Figs. 2 and 3) to consolidate this conclusion.

Figure 1.6 FTIR spectra of Met and MFNs.

5. Reusability and Stability: The manuscript reports on the stability and reusability of the MFNs, but this section would benefit from more comprehensive data on the long-term performance of the material. Including insights into potential degradation mechanisms or factors that could affect long-term stability would provide more depth to this analysis. While the results are promising, a more thorough comparison with alternative materials, both in terms of performance and cost, would give readers a better understanding of the potential impact of this method.

- Physical and chemical property analyses of the materials before and after the recycling test have been conducted to investigate the sensitive factors for long-term stability and the related impact mechanisms. XRD analysis of MFNs before and after recycling test was performed to explore the effect of continuous CO_2 capture on the structural stability of material. As shown in Figure 1.7a, the diffraction peak positions in the XRD patterns of the two samples are completely consistent, indicating that the crystalline phase structure of Fe_3O_4 remains stable during the recycling test, and the amorphous Met did not crystallize. Fe 2p spectrum (Figure 1.7d) indicates that the ratio of Fe^{2+} to Fe^{3+} peak areas of the recycled MFNs was similar to that of fresh MFNs, suggesting that no significant oxidation or reduction of Fe_3O_4 occurred during the recycling test. The comparison of FTIR spectra of the samples before and after recycling test (Figure 1.7b)

shows that the characteristic peak positions, intensities, and shapes of the Fe-O bonds of Fe_3O_4 and the functional groups (C=O, C-O, O-H) of Met remain consistent, implying that the surface Met did not degrade, decompose, or participate in chemical reactions. The C1s spectra of the two samples are almost identical (Figure 1.7e), indicating that the C-N, C-O, C-S, and C=O groups derived from Met did not change significantly, confirming that the chemical structure of Met was not destroyed or decomposed. O1s spectra show that the lattice oxygen ratio of MFNs after recycling test slightly increased, and the signal of surface adsorbed oxygen species at 531.2 eV slightly weakened, indicating a minor desorption of Met from the surface. The TGA curve indicates that the weight loss of MFNs after recycling test was 2.84%, which quantified the Met desorption content on the surface of material (Figure 1.7c). Compared to bare Fe_3O_4 , the remaining Met loading of MFNs after recycling was 9.07%, confirming that the performance of the recycled MFNs still surpasses that of Fe_3O_4 nanoparticles. In summary, the structure and surface chemical state of the samples remained stable during recycling test, with no significant changes in the crystalline phase and chemical state of Fe_3O_4 . The structure and binding pattern of the surface-loaded Met remained intact, with only a minor desorption occurring. Related detailed characterization information has been added to Supplementary Fig. 14.

Figure 1.7 Comparison of physicochemical properties of MFNs before and after recycling test. (a) XRD patterns. (b) FT-IR spectra. (c) TG curves. (d-f) High-resolution XPS spectra of Fe 2p (d), C 1s (e), and O 1s (f) for fresh and recycled MFNs.

Sustainable CO_2 capture cycle based on hydrogen-bonded water cages involves a variety of physical and chemical interactions, such as thermodynamic changes associated with crystallization and decomposition, mechanical disturbances, hydrodynamic forces, local stress changes due to crystallization, and the migration of dissociated CO_2 gas. The main operating temperature ranges from 275 K to 298 K, with CO_2 introduction at 288 K, followed by temperature adjustment to 275 K to induce hydrogen-bonded water cages formation and CO_2 capture, and CO_2 @Water crystals were decomposed at room temperature (298 K) for MFNs/water regeneration after the capture process. The temperature rise caused by exothermic crystallization was ~ 1 K, which did not lead to significant changes in the thermodynamic conditions of the solution. Within the temperature range, the physical and

chemical properties of MFNs remain stable, as reflected in the TG curves of fresh samples and recycled samples, indicating that operational temperature changes and thermal flow fluctuations caused by crystallization were not the main factors affecting material stability. To enhance CO₂ diffusion and induce crystallization, the solution was perturbed at a stirring rate of 100 r/min, which inevitably resulted in some loss in the Met loading due to friction between MFNs and stirrer and reactor wall. The shear forces of fluid motion enhanced by stirring and the collision between nanoparticles also increased the possibility of Met shedding. The crystallization process of liquid water transforming into hydrogen-bonded water cages and capturing CO₂ is accompanied by volume expansion (~10%), which will exert mechanical compression and shear stress on the surrounding MFNs, leading to compression and tearing of Met layer and causing partial detachment. At the end of CO₂ trapping process, MFNs are dispersed in the porous solid formed by the co-crystallization of hydrogen-bonded water cages with CO₂. To achieve reversible CO₂ release and solution regeneration, the crystals are placed under ambient temperature for decomposition. As the local crystals collapse and evolve, Met detachment from MFNs will be exacerbated by the seepage of dissociated gas within pores and cracks and the fluid erosion caused by dissociated water migration. Therefore, the promotion performance of MFNs is weakened with repeated cycles of CO₂ trapping in hydrogen-bonded water cages, which is attributed to Met shedding influenced by various mechanical effects during crystallization and decomposition.

The cost of MFNs was estimated based on the cost of reagents and compared with that of the alternative material sodium dodecyl sulfate (SDS) in terms of cost and CO₂ capture performance (Table 1.1). The cost of MFNs is approximately five times that of SDS; and MFNs have 1.5 times CO₂ capture capacity and 43% t₉₀ of SDS. Although MFNs are not cost-competitive initially, their excellent recyclability and promoting performance make them advantageous for long-term and continuous CO₂ capture process. After 17 recycling cycles, the CO₂ capture capacity of MFNs remains at 109.37 v/v, which is 38.4% higher than that of SDS. Furthermore, during the CO₂@Water crystal decomposition process, SDS tends to produce foam and mixes into the gas stream, resulting in resource loss and potential water purification costs. This indicates that MFNs with high-performance stability, are suitable for long-term sustainable CO₂ capture, with their economic advantages becoming more pronounced over extended recycling periods. Cost and performance comparisons with alternative materials were added to the Supplemental Information (Supplementary Table 4).

Table 1.1 Comparison of Costs and CO₂ Capture capacity Between MFNs and SDS.

Material	Reagent	Purity	Reagent cost* (\$/g)	Cost (\$/g)	CO ₂ capture capacity (v/v)	t ₉₀ (min)
SDS	-	97%	0.0118	0.0118	79.0 ± 6.2	820.4 ± 98.3
	L-methionine	98%	0.0188	0.0602		
MFNs	FeCl ₃ ·6H ₂ O	90%	0.0033		117.5 ± 2.1	352.9 ± 33.9
	FeSO ₄ ·7H ₂ O	90%	0.0035			
	NaOH	95%	0.0032			

*The reagent cost comes from Shanghai Macklin Biochemical Technology Co., Ltd.

6. Literature Review: The literature review could be strengthened by including more recent advancements in CO₂ capture technologies. Comparing this electrochemical approach with other emerging methods, such as those using porous carbons or zeolites, would provide a more comprehensive view of the field.

- *Many thanks for this valuable comment, which is important for the improvement of the literature review.*

Currently, emerging CO₂ capture methods are developing rapidly with the goal of capturing CO₂ in a more efficient, cost-effective, and environmentally friendly manner. Some representative CO₂ capture technologies have attracted widespread research interest, such as physical adsorption using porous materials, chemical adsorption by solid amine-based materials, carbonate crystallization, and bio-based CO₂ capture technologies. Porous material adsorption is known for its efficiency, flexibility, and economy, and has attracted significant attention in CO₂ capture technology. Developing porous materials with high specific surface areas, adjustable pore sizes, and selective adsorption properties is key to improving CO₂ capture efficiency. The large surface area and pore volume, high adsorption selectivity, excellent chemical and thermal stability make porous carbon and zeolites highly promising in the field of CO₂ adsorption. Some optimization strategies including the design of high-performance porous structures, surface amine modification, and enhancing adaptability to humid conditions, have been focused to further improve the capture performance of these porous materials under practical conditions. For chemical adsorption, solid amine-based adsorbents are gradually replacing liquid amine absorption to reduce solvent evaporation, sharing advantages such as high adsorption capacity, tunable structure, and stable adsorption performance, but are plagued by amine degradation and high energy consumption for regeneration. Carbonate crystallization can fix CO₂ efficiently, but the high energy consumption for CO₂ regeneration and the increased economic cost due to the non-recyclability of the absorbent are prohibitive. In the revised manuscript, these recent advances on emerging technologies for CO₂ capture have been supplemented and fully compared with the methodology proposed in this study.

7. Figures and Tables: The figures are informative but could benefit from more descriptive captions. Including key experimental details or takeaways in the captions would improve clarity. A comparative table summarizing the performance of various CO₂ capture technologies discussed in the manuscript would be a useful addition.

- *Thanks for Your kind reminder. In the revised manuscript, the figure captions have been supplemented with key details and changed to be more descriptive to clearly present the contents of the figures. The manuscript discusses several representative CO₂ capture technologies, including liquid amine absorption, porous material adsorption, and carbonate crystallization, and clear performance comparisons between these technologies will make the manuscript more intuitive and comprehensive. A table has been added to systematically compare the performance indexes of several representative CO₂ capture technologies mentioned in the manuscript, including advantages, regeneration temperatures, adaptation conditions and environmental impacts (Supplementary Table 2). This table will help readers in gaining a clearer understanding of the advantages and disadvantages of different CO₂ capture technologies, as well as the applicable scenarios.*

Supplementary Table 2

Comprehensive comparison of the advantages, regeneration temperatures, adaptation conditions and environmental impacts of several representative CO₂ capture methods.

Method	Advantages	Regeneration temperature	Environmental impacts	Adaptation conditions	Ref.
Liquid amine absorption	Capture efficiency, selectivity, and device compatibility	>100 °C	Corrosive equipment, generation of degradation products	High CO ₂ concentration	16
Porous material adsorption	Flexibility and recyclability	Spent zeolites: 250-300 °C Activated carbon: 70-200 °C	No solvent contamination, recyclable materials	Low CO ₂ concentration	17, 18
Carbonate crystallization	Capture efficiency and stable product	CaCO ₃ : >700 °C	Strong corrosiveness of alkaline solutions	High CO ₂ concentration	19
Hydrogen-bonded water cage	Capture capacity, insensitivity to impurities in industrial gases	Ambient (25 °C)	Water as a crystallization precursor, green reaction	High CO ₂ concentration	20

8. References: There are some formatting inconsistencies in the references that should be addressed. Ensuring uniformity in the reference style would enhance the professionalism of the manuscript.

- *Thank you for your valuable comments. In the revised manuscript, we have carefully checked and standardised the format of the references to ensure compliance and made necessary adjustments to each reference to improve the professionalism and standardization of the manuscript.*

Conclusion:

While this manuscript presents an interesting approach to CO₂ capture, there are several significant areas that need improvement, particularly in terms of clarifying the CO₂ adsorption mechanism, enhancing material characterization, and providing more detailed experimental data. Additionally, a more comprehensive comparison with current CO₂ capture technologies would help to contextualize the findings.

- *Many thanks for the careful review and valuable comments, which are important for the improvement of this work. Based on these comments, we have carefully revised the manuscript: molecular dynamics simulations and in-situ Raman spectroscopy were performed to illustrate the mechanism of hydrogen-bonded water cages formation and CO₂ capture; XRD, TEM, FTIR, TGA and XPS analysis of the materials and recycled materials were added to illustrate the material structure and characteristics; specific details of the experimental procedure were added and the performance of the designed materials were compared with the alternatives to improve the details of the study; a more comprehensive comparison with representative CO₂ capture technologies in the advantages, regeneration temperatures, adaptation conditions and environmental impacts, etc. is provided to clearly introduce the related research progress in this field.*

Reviewer #2 (Remarks to the Author):

The manuscript presents an innovative and potentially impactful approach to carbon capture using a crystallization-based method facilitated by Methionine@Fe₃O₄ nanoparticles. This strategy enhances the CO₂ capture process by inducing hydrogen-bonded water nanocages, which demonstrates significant improvements over traditional crystallization-based CO₂ capture methods. The use of magnetic nanoparticles with biocompatibility and recyclability characteristics positions this work as a novel contribution to the field of sustainable CO₂ capture technology. However, while the research is of high quality and offers substantial promise, a few minor revisions are necessary before publication.

- *Many thanks for the recognition and valuable comments on this work. Based on these comments, we have carefully revised the manuscript, including revising the textual presentation, improving the material characterization, supplementing the experimental data and analyses, and strengthening the mechanistic analyses. We believe that these comments have significantly improved the quality of this manuscript.*

1. Some terms in the supplementary equations lack unit definitions. Appropriate units should be included for all variables.

- *Thank you very much for your kind reminder. In the revised supplementary information, we have added unit definitions for the terms mentioned, and units have been added for all variables.*

2. The statement in the manuscript, line 140, regarding the EDS elemental mapping images as well as their overlaps (Fig. 2e and Supplementary Fig. 1) revealed the homogeneous distribution of Fe, O, N, C, and S elements..." is not fully supported by the data presented. The core-shell structure and particle size (~75 nm) of MFNs are not clearly visible in Figure 2e. Higher-resolution imaging is needed for validation.

- *Thanks for this valuable comment. In the revised Supplementary Information, the TEM images of MFNs as well as the overlapping images of TEM images with elemental mapping images of relevant elements were added to Supplementary Figure. 1 to clearly illustrate the material properties, as shown in Figures. 2.1a-c below. Figure 2.1c clearly shows that all elemental mappings are centrally distributed on the sample and homogeneously dispersed. Fe elemental signals are masked by the signals of elements including O, C, N, and S, which confirms the effective encapsulation of Fe₃O₄ nanoparticles by Met. In addition, TEM images of other regions of MFNs and overlapping images of associated elements were supplemented to consolidate this conclusion, as shown in Figure 2.1d-f. The elements in this region are also concentrated and uniformly dispersed on the surface of the samples, and the signal of elemental mapping derived from Met such as O, C, N, and S is obvious, which proves the structure of Met-coated Fe₃O₄. Two characterization data are consistent and both confirm the uniform distribution of elements in MFNs and the effective encapsulation of Fe₃O₄ nanoparticles by Met.*

Figure. 2.1 (a, d) TEM images of MFNs in different regions, (b, e) overlap of elemental mappings of Fe, O, C, N, and S, and (c, f) TEM images with all elemental mappings.

Higher-resolution TEM images were added to the revised manuscript (Fig. 2c, d) to show the structure of MFNs clearly. The TEM image (Figure 2.2a) shows that the particle sizes of MFNs are predominantly distributed in the range of 40-70 nm, with a median size of 54.93 nm. As shown in Figure. 2.2b, MFNs exhibit the structure of Met layer encapsulating Fe_3O_4 core. The amorphous Met exists as a light-colored layer with distinct edges and a clear boundary with the Fe_3O_4 nucleus. The lattice fringes of Fe_3O_4 nucleus confirms its crystalline phase (Figure 2.2c), the thickness of Met layer ranges from 12.6 to 13.2 nm, and this detail has been added to the revised manuscript to elaborate on the material information.

Figure 2.2 TEM image (a) and HRTEM images (b) and (c) of MFNs.

3. In Figure 3, which shows the "Pressure and temperature evolution during CO_2 @Water clathrate formation,". The X-axis of Figure 3 should include "Time (min)" to ensure proper interpretation of the figure.

- Thank you very much for your attention to this detail. The X-axis of Fig. 3 represents the reaction time in minutes. The X-axis labelling of Fig. 3 (Page 14) has been added in the revised manuscript to achieve better clarity.

4. For recycling performance of the MFNs-0.5, were the results shown in Figure 6 reproducible? Did each sample fail at the same point and in the same way? How long did it take for the average nanoparticles to fail? Provide details on the number of trials and statistical analysis for the recycling performance in Figure 6 to confirm the robustness of the results.

- *Many thanks for your interest in the recycling performance of the materials and valuable comments. To ensure the reliability and robustness of the results, three parallel experiments were carried out for the recycling performance tests, and the results of the three sets of experiments were statistically analysed to assess the CO₂ capture capacity and the reaction kinetics parameter (t_{90}). These data and the corresponding errors are presented in the revised manuscript (Fig. 6, Page 25). The ultimate CO₂ capture capacity and reaction kinetics are main focus in the recycling performance tests of this study. Figure 2.3b shows the ultimate CO₂ capture capacity obtained for each cycle, and the results show excellent reproducibility, with relative errors ranging from 0.05% to 6.1%. Due to the inherent stochastic property of crystalline nucleation, CO₂ capture based on hydrogen-bonded water cages tends to exhibit inconsistent trends in reaction kinetics. In this study, MFNs-0.5 has significantly attenuated the stochastic property of crystal nucleation and growth in liquid water, and the CO₂ capture kinetics of different experimental groups exhibit similar trends, although they are not identical, as reflected in the time-varying CO₂ capture curves with error bands in Figure 2.3a. To clearly assess the recycling performance of the materials and to guarantee the reliability of the results, the recycling tests were supplemented from the initial 8 to 17 and were carried out in three parallel sets. CO₂ capture capacity was not significantly weakened with repeated cycling, and the ultimate CO₂ capture capacity remained at 109.37 v/v, which implies the excellent recycling performance of MFNs. The reaction kinetics showed a gradual weakening trend with the increase of cycling times, although it still outperformed the comparison material (SDS), as shown in Figure 2.3c. Mechanical perturbations during continuous cycling tests, mechanical effects from crystalline expansion with dissociative CO₂/water multiphase percolation increased the risk of Met shedding from MFNs surface, which in turn weakened the promotional performance of MFNs. The initial materials, reactors, temperature and pressure conditions, and operating conditions used in the three parallel experiments were identical. The displayed trends in reaction kinetics with increasing cycle number were consistent across all groups, with the relative error in the content of CO₂@Water crystals within 6.1%, indicating that implying that the materials were subjected to almost identical mechanical effects during the cycling tests. Thermogravimetric analysis (TGA) of two MFNs samples after cycling testing (Figure 2.4) showed consistent weight loss trends, with the difference in final weight loss being within 0.5%, implying that MFNs failed in the same mode. TGA data of recycled materials was supplemented to the revised Supplementary Information, and a discussion on the failure mode was added to the revised manuscript. Although the ultimate CO₂ capture capacity showed that MFNs maintained excellent promotion performance with continuous cyclic reactions, the reaction kinetics gradually deteriorated. The failure time of material was defined from reaction kinetics, where the average t_{90} exceeding 90% of the reaction time was considered as material failure. In this study, after 17 cycles of testing, t_{90} reached 631 min, exceeding 90% of the total reaction time (700 min), thus determining that the material failed during this cycle, with the failure time being the sum of the reaction times of the first 16 cycles (11200 min). In the revised manuscript (Pages 24 and 25), the number of tests and statistical analysis were supplemented in detail, and the definition of material failure point and the corresponding failure time were clearly explained.*

Figure 2.3 Recycling performance of MFNs-0.5 in 17 individual recycling experiments. (a) Time-dependent CO_2 capture curves. (b) Ultimate CO_2 capture capacity. (c) t_{90} .

Figure 2.4 Comparison of TG curves of two MFNs tested over 17 cycles.

5. The authors suggest that the proposed method holds great potential for industrial-scale CO_2 capture, but there is little discussion on how this would be achieved practically. For example: The claim of phase transition energy for CO_2 capture using MFNs is as low as 0.57 kJ/mol CO_2 , which is indeed lower than

other methods. However, a detailed energy balance for the entire capture process, including cooling, crystallization, and regeneration, is required to evaluate the industrial feasibility of this method.

- *We apologize for this misleading description, the enthalpy change of CO₂@Water crystal is 57.1 kJ/mol, which has been corrected. As shown in Fig. 8a, the enthalpy change of CO₂@Water phase transition remains advantageous compared to representative CO₂ absorbers. And we thank the reviewer for raising this important point regarding the practicality and industrial feasibility of our proposed method. To further address the concern, we have expanded the discussion in the manuscript to provide additional context on the energy balance and practical considerations for industrial implementation. Specifically, we consider the energy requirements for three major steps: cooling, crystallization, and regeneration. The key points are as follows: Cooling: The cooling requirement for the water-MFN dispersion is minimized by utilizing efficient thermal management strategies, such as industrial waste heat recovery or integration with ambient cooling systems. These measures substantially reduce the energy input required for cooling to the target temperature (275 K), making the process more energy efficient in industrial settings.*

Crystallization: The crystallization process is driven by the formation of hydrogen-bonded water nanocages, which occurs spontaneously under the specified conditions.

Regeneration: A key advantage of our system is the simple, low-energy regeneration of MFNs, which occurs through ambient heat exchange at room temperature. Unlike traditional sorbents that require high-temperature desorption, the MFNs remain stable and effective for reuse without energy-intensive processing. This feature further enhances the energy efficiency and sustainability of the process.

To provide greater clarity, the following paragraph has been added to the manuscript:

"For the practicality and industrial feasibility, cooling energy demands can be reduced by leveraging industrial waste heat recovery or ambient cooling in appropriate settings. Some gas guest (such as ethane, propane and isobutane) from industrial sources can be used as additives to increase the operating temperature by weakening crystallization energy barrier to alleviate energy demand. Moreover, the regeneration of MFNs is achieved through a simple room-temperature heat exchange, eliminating the requirement for energy-intensive desorption processes. These features highlight the feasibility and sustainability of the proposed method, particularly for industrial applications where low-grade heat sources are available."

We acknowledge that a detailed energy balance, including a techno-economic analysis, is critical for assessing the industrial scalability of this method. However, given the early-stage nature of this work, such an analysis requires further experimental and modeling efforts, which are part of our ongoing and future research. This limitation is now explicitly mentioned in the revised manuscript.

We hope that this expanded discussion adequately addresses your concerns regarding the practical implementation and feasibility of the proposed method. Please let us know if further clarification or additional analysis is required.

6. The manuscript mentions that increasing MFNs concentration beyond 0.5 wt% led to reduced performance due to agglomeration and viscosity effects. However, this is only briefly discussed and needs a more in-depth exploration. If agglomeration of nanoparticles is a limiting factor in this system, it could hinder its scalability. Have you considered methods to prevent nanoparticle agglomeration, and what are the potential trade-offs between preventing agglomeration and maintaining CO₂ capture efficiency?

- Thank you for this valuable comment. Outstanding CO₂ capture performance relies on the concentration-dependent Met abundance and their dispersion in the solution. To elucidate the dependence of MFNs dispersion on the concentration, the zeta potential and dynamic light scattering (DLS) hydrodynamic diameter of nanoparticles at different concentrations were determined using a Zetasizer Nano ZS90 (Malvern, UK). As shown in Figure 2.5a, the absolute value of zeta potential decreased from 20.2 mV to 10.8 mV as the concentration of MFNs was increased from 0.05 wt% to 0.5 wt%; however, as the concentration was further increased to 1.0 wt%, it was significantly decreased to 3.07 mV. This indicates that the enhanced interactions between MFNs particles decrease the stability of nanoparticles in solution with increasing concentration. The hydrodynamic diameter of MFNs at different concentrations are illustrated in Figures 2.5b-f, which show that the particle diameters ranged from 375.05-834.12 nm in the concentration range of 0.05-0.5 wt%; however, the particle hydrodynamic diameter increased significantly (up to 1603.89 nm) with a further increase in the concentration (1.0 wt%), suggesting the significant agglomeration of MFNs.

Figure 2.5 The dependence of MFNs stability on the concentration. (a) Zeta potential of MFNs at different concentrations. (b)-(f) Hydrodynamic diameter of MFNs at 0.05, 0.1, 0.2, 0.5, and 1.0 wt% concentrations, respectively.

Some small molecular surfactants and water-soluble polymers can effectively prevent particle aggregation by adsorbing on the surface of nanoparticles to form electrostatic repulsion, steric hindrance, or hydration layers. However, the high toxicity and poor biodegradability of chemical surfactants such as sodium dodecyl sulfate (SDS), cetyltrimethylammonium bromide (CTAB), and polyoxyethylene alkyl phenol ether (Triton X-100) pose potential environmental risks and are not suitable for sustainable green CO₂ capture processes. Although biodegradable natural polymers like guar gum and chitosan provide an environmentally friendly alternative for nanoparticle dispersion, the associated high viscosity of liquid and disordered water molecule arrangement inhibit CO₂ diffusion and increase the energy barrier for crystal nucleation, which is detrimental to the reaction kinetics of CO₂ capture. Moreover, the use of water-soluble dispersants increases the difficulty of liquid water purification and recycling, and their impact on environmental and economic costs cannot be ignored. The abundance and dispersion of Met

significantly depend on the concentration of nanoparticles, which directly affects the CO₂ capture capacity and reaction kinetics. MFNs in the concentration range of 0.2-1.0 wt% achieved excellent CO₂ capture capacities (114.0-118.7 v/v), but the reaction kinetics were mutually constrained by nanoparticle dispersion and the promoter abundance. The promoter abundance of 0.5 wt% MFNs is 2.5 times higher than that of 0.2 wt%, with a similar difference of dispersion (as reflected by zeta potential values), which ensured the superior performance of MFNs-0.5. The weakened promotional performance of MFNs-1.0 can be attributed to the high tendency of aggregation, despite possessing a high promoter abundance. An effective promoter abundance and the prevention of nanoparticle aggregation are necessary conditions for excellent CO₂ capture performance, which can be achieved by regulating the concentration of MFNs. In addition, this strategy can be achieved by relying only on modulating MFNs dose, and the material is magnetically recyclable, avoiding the use of environmentally unfriendly and difficult-to-separate chemical dispersants to raise environmental risks and economic costs.

7. The explanation of how Methionine@Fe₃O₄ nanoparticles induce local water ordering and enhance CO₂ capture kinetics needs more detailed mechanistic insights. Although the manuscript provides some discussion around hydrogen bonding and micro-convection, further experimental evidence (such as fluid dynamics data) or theoretical modelling would help to support these claims.

- *Many thanks for your valuable suggestion. Experimental evidence for improved solution heat and mass transfer performance of Methionine@Fe₃O₄ nanoparticles (MFNs) have been added in Supplementary information (Supplementary Fig. 8) with microscopic validation of their enhanced CO₂ capture kinetics. Hydrogen-bonded water cages-based CO₂ capture process includes CO₂ dissolution, co-crystallization of CO₂ with water, and subsequent continuous crystal growth, which are dependent on the heat and mass transfer performance of fluid. Raman spectroscopy measurements of the stretching vibrations of water have revealed that MFNs can promote the orderly arrangement of water molecules through hydrogen bonding, laying the foundation for the subsequent formation and growth of water cages. Additionally, the co-crystallization of CO₂@Water is an exothermic reaction, and the rapid diffusion of latent heat can effectively maintain high temperature gradient to ensure continuous CO₂ capture, which primarily relies on the thermal conductivity of fluid. The thermal diffusion coefficient and thermal conductivity of pure water and 0.5 wt% MFNs dispersion at ambient temperature (298 K) and experimental temperature (275 K) were measured using a Differential Scanning Calorimeter (DSC 300, NETZSCH, Germany) and a Thermal Conductivity Analyzer (LFA467, NETZSCH, Germany) to analyze the impact of MFNs on the heat and mass transfer of fluid. As shown in Figure 2.6a and d, the thermal diffusion coefficient and thermal conductivity of the MFNs dispersion were higher than those of pure water at both ambient temperature and experimental temperature. At the experimental temperature (275 K), MFNs increased the thermal conductivity of water by 14.5%, even exceeding the heat conduction capacity of pure water at ambient temperature, indicating the excellent ability of MFNs for the improvement of heat and mass transfer property of fluid. A high pressure differential scanning calorimeter (HP μ -DSC7 Evo, Setaram) was employed to in-situ determine the effect of MFNs on the kinetic performance of CO₂ capture by monitoring the heat flow changes (Figure 2.6b and e). The reaction was carried out at a constant pressure of 4 MPa with an initial temperature of 288 K. The temperature was reduced to the experimental temperature (275 K) at a rate of 0.5 K/min and then maintained constant, with a fluid mass of 30 mg for a total reaction time of 10 h. No heat flow release was detected in the pure water*

system throughout the reaction period, implying that water molecules did not form hydrogen-bonded cages to trap CO_2 , which existed in the free gas state. The presence of MFNs significantly improved the nucleation kinetics of hydrogen-bonded water cages, which was reflected in the pronounced peak of released heat flow and the short induction time for nucleation. In addition, the influencing mechanism of MFNs on the growth kinetics of hydrogen-bonded water cages was probed in-situ by a LabRAM HR evolution confocal Raman spectrometer equipped with an 1800 grooves/mm grating. The laser excitation wavelength was 532 nm, output power was 100 mW, acquisition range was 1200 to 1500 cm^{-1} , and the spectra was acquired three times with exposure time of 60 s. A capillary optical cell with an inner diameter of 300 μm was placed on a high-pressure cooling table as the reactor. The liquid amount for reaction was 1 μL , and the experimental conditions were kept constant at 4 MPa and 275 K. CO_2 gas signal was initially scanned as a reference, followed by microscopic observation of liquid state in the capillary cell. And once CO_2 @Water crystals were detected, the position was immediately focused and Raman spectra were continuously acquired at 5 min intervals, with a reaction time of 45 min. As shown in Figure 2.6c, the two peaks at 1286.5 and 1389.3 cm^{-1} were corresponded to CO_2 gas, the signals of CO_2 trapped by hydrogen-bonded water cages were shifted to 1276.3 and 1383.8 cm^{-1} in pure water, respectively. And the peaks of CO_2 captured in the MFNs dispersion located at 1275.9 and 1384.5 cm^{-1} , respectively (Figure 2.6f), which were almost the same as that of pure water, indicating the hydrogen-bonded water cages were not destroyed by MFNs. In addition, the time-varying Raman spectra showed that CO_2 capture in pure water stagnated after 25 min, which was reflected in the consistent peak intensity of Raman spectra within 25-45 min; the signals of captured CO_2 in MFNs system was continuously enhanced, which implied that CO_2 was consistently captured into the hydrogen-bonded water cages with maintained reaction kinetics. The excellent crystal nucleation and growth kinetics imply that the heat and mass transfer property of fluid is significantly improved by MFNs, which facilitates the scale-up generation of hydrogen-bonded water cages. Relevant experimental evidence has been added to the revised manuscript and supplementary information to illustrate the promotional performance and influencing mechanism of MFNs on CO_2 capture by hydrogen-bonded water cages.

Figure 2.6 Thermal diffusion coefficients and thermal conductivities of pure water (a) and 0.5 wt% MFNs dispersion (d) at ambient temperature and experimental temperature, respectively. Temperature profiles and heat flow changes of pure water (b) and 0.5 wt% MFNs dispersion (e) monitored by in-situ DSC, respectively. In-situ time-varying Raman spectra during CO₂ trapping in hydrogen-bonded water cages of pure water (c) and 0.5 wt% MFNs dispersion (f), respectively.

Reviewer #3 (Remarks to the Author):

In this paper, Wang et al. presented organic magnetic nanoparticles, (Methionine@Fe₃O₄, termed as MFNs), which trapped CO₂ via pressure swing crystallization with a high capacity (118.7 v/v) and density (22.7 wt%), thus providing a new inspiration for sustainable CO₂ capture with zero resource depletion (ZRD). In addition, MFNs have good biocompatibility, which provides a safety guarantee for their application and promotion. The specific design concept of this work is novel and unique. The manuscript provides valuable insights into the development of future sustainable CO₂ capture and storage. I agree that this manuscript to be accepted after some minor revision. The following are the suggestions for the authors to improve the manuscript.

- *Many thanks for your recognition and valuable comments. Careful revisions have been made based on these comments, which have greatly contributed to improving the quality of the manuscript.*

1. The aggregation problem of magnetic nanoparticles can lead to a decline in material performance and affect the dispersibility in water. Discuss whether it has a certain impact on CO₂ capture performance.

- *Thank you very much for this valuable comment. Aggregation of magnetic nanoparticles affects their dispersibility in water, which in turn affects CO₂ trapping performance. To elucidate the aggregation of magnetic nanoparticles and the dependence of dispersibility on concentration, the Zetasizer Nano ZS90 (Malvern, UK) was applied to determine the hydrodynamic diameter and zeta potential of nanoparticles by dynamic light scattering (DLS). As shown in Figure 3.1, the median hydrodynamic diameter of nanoparticles raised from 375.05 nm to 834.12 nm as the concentration increased from 0.05 wt% to 0.5 wt%; in particular, the median hydrodynamic diameter of magnetic nanoparticles in the 1.0 wt% system reached 1603.89 nm. This implies that the agglomeration of magnetic nanoparticles intensified with increasing concentration, especially nanoparticles showed significant agglomeration at high concentrations (1.0 wt%). This statement is consolidated by zeta potential of magnetic nanoparticles with different concentrations. As shown in Figure 3.2a, the absolute value of zeta potential decreased with increasing concentration of magnetic nanoparticles, indicating that the dispersion of material was gradually weakened, especially the aggregation of magnetic nanoparticles at high concentration (1.0 wt%) resulted in the worst dispersibility. The CO₂ trapping performance is strongly dependent on the abundance of promoter and their dispersion in the solution. Despite high concentrated (1.0 wt%) MFNs dispersion possessing considerable promoter abundance, however, their weak dispersion limited the performance, which was manifested in the relatively*

poor CO₂ capture kinetics (Figure 3.2b). Affected by the mutual constraints of nanoparticle concentration and dispersibility in solution, 0.5 wt% MFNs present the most favorable CO₂ trapping kinetics. The effect of magnetic nanoparticle aggregation and dispersibility on CO₂ capture performance has been elaborated in the revised manuscript (Page 22-23), and the relevant detailed characterization has been added to the Supplementary Information (Supplementary Fig. 11) for consolidation.

Figure 3.1 Hydrodynamic particle size distribution of MFNs at (a) 0.05 wt%, (b) 0.1 wt%, (c) 0.2 wt%, (d) 0.5 wt%, and (e) 1.0 wt% concentrations. (f) Mean hydrodynamic diameter of MFNs at different concentrations.

Figure 3.2 (a) Zeta potentials of MFNs at different concentrations. (b) Time-varying CO₂ capture curves in 0.2, 0.5, and 1.0 wt% MFNs dispersions.

2. XRD, XPS, and EDS etc. abbreviations should be expanded in full for the first time they appear in the text and then used in abbreviated form thereafter. For example, XPS is expanded in line 173 and EDS in line 503, rather than being expanded for the first time they appear.

- We have checked the manuscript and ensured that all acronyms (e.g., XRD, XPS, and EDS, etc.) have been expanded to their full names on their first appearance, and the abbreviated forms

are used consistently in subsequent content. Thanks for your correction, which helps to improve the standardization of the manuscript.

3. References should be added to support the conclusions of lines 101-103 in manuscript.

- *Thanks for your kind reminder. Relevant reference has been added here to consolidate this conclusion: "Surfactants and nanomaterials have been proven to overcome barriers of gas-liquid diffusion and crystal nucleation by improving heat and mass transfer conditions."⁴⁰*

4. The picture definition is low. Please check the full text.

- *In the revised manuscript, the images have been reprocessed to improve resolution and contrast to ensure that the details of the images are clearer.*